# Training Compute-Optimal Protein Language Models

**Xingyi Cheng**[1*]**, Bo Chen**[2*†]**, Pan Li**[1]**, Jing Gong**[1]**, Jie Tang**[2]**, Le Song**[1,3]
[1]BioMap Research [2]Tsinghua University [3]MBZUAI
derrickzy@gmail.com,cb21@mails.tsinghua.edu.cn
Code: https://github.com/cxysteven/ScalingProteinLM

## Abstract

We explore optimally training protein language models, an area of significant interest in biological research where guidance on best practices is limited. Most models are trained with extensive compute resources until performance gains plateau, focusing primarily on increasing model sizes rather than optimizing the efficient compute frontier that balances performance and compute budgets. Our investigation is grounded in a massive dataset consisting of 939 million protein sequences. We trained over 300 models ranging from 3.5 million to 10.7 billion parameters on 5 to 200 billion unique tokens, to investigate the relations between model sizes, training token numbers, and objectives. First, we observed the effect of diminishing returns for the Causal Language Model (CLM) and that of overfitting for the Masked Language Model (MLM) when repeating the commonly used Uniref database. To address this, we included metagenomic protein sequences in the training set to increase the diversity and avoid the plateau or overfitting effects. Second, we obtained the scaling laws of CLM and MLM on Transformer, tailored to the specific characteristics of protein sequence data. Third, we observe a transfer scaling phenomenon from CLM to MLM, further demonstrating the effectiveness of transfer through scaling behaviors based on estimated Effectively Transferred Tokens. Finally, to validate our scaling laws, we compare the large-scale versions of ESM-2 and PROGEN2 on downstream tasks, encompassing evaluations of protein generation as well as structure- and function-related tasks, all within less or equivalent pre-training compute budgets.

## 1 Introduction

Scaling up transformer-based models has become a guiding principle for enhancing model performance across broad domains, particularly in Natural Language Processing (NLP) [4, 12, 24, 64, 79] and Computer Vision (CV) [20, 67, 89]. In recent years, large transformer-based Protein Language Models (PLMs) such as PROGEN familiy [51, 61], ESM familiy [68, 47] and xTrimoPGLM [14] have also been developed, which leads to significant improvements over model performance on complex downstream tasks [27, 45]. Current language models utilize two main training objectives to encode sequence information: the BERT-like [23] Masked Language Model (MLM) and the GPT-like Causal Language Model (CLM) [11], each applied either separately or in a unified fashion. A common understanding is that bi-directionally MLM excels in sample efficiency and shows enhanced performance in downstream task fine-tuning. This is particularly true in tasks that emphasize understanding complex patterns, making it a prevalent learning objective in modeling protein sequences[*]

---

[*]XC and BC contributed equally.
[†]Work done while interned at BioMap.
[*]Appendix D also compared CLM and MLM on the protein contact prediction task through fine-tuning and freeze probing, with MLM demonstrating superior performance relative to CLM.

38th Conference on Neural Information Processing Systems (NeurIPS 2024).

[46, 14]. On the other hand, uni-directional CLM, due to its sequential generation ability, is better suited for generating more coherent and realistic sequences compared to MLM [19, 61, 63].

However, training large protein language models (PLMs) are computational-intensive, and strategies for optimally allocating compute budgets for training PLMs are relatively underexplored, with *most efforts focusing on scaling model parameters based on a fixed set of training tokens to achieve performance improvements*. A key insight [36, 42, 76] is that large models should not be trained to their lowest possible loss to optimize computing; instead, models and data should be scaled proportionally based on available compute budgets. These scaling laws are broadly found in natural language models [42, 36, 33, 2, 58, 78, 17, 90]. But their applicability has not been validated within biological datasets, such as the primary structures of proteins, which are composed of amino acid sequences forming protein chains. Unlike natural languages, protein sequences are scientific data that are precisely represented using a vocabulary of 20 amino acids, with very little redundancy and are not as semantically smooth. Thus, we consider such data as a distinct modality and ask the question: *What are the scaling behaviors for MLM and CLM in protein language modeling?*

We focus on the best practices, which include revisiting datasets, optimization objectives, and model parameters as key factors. Our goal is to investigate an optimal training scheme for protein language models given predetermined compute budgets. Our core findings are as follows:

- We revisited the protein sequence data used for training PLMs and collected a dataset of 194 billion unique tokens on 939M unique sequences from publicly available sources to address the issue of overfitting and perform plateau in protein language modeling.

- We find that, in both MLM and CLM, training data scales sublinearly in the model sizes but follow distinct power-laws. In other words, a $10\times$ increase in compute leads to a $6\times$ increase in MLM model size and a 70% increase in data, versus a $4\times$ increase in CLM model size and a $3\times$ increase in training tokens.

- We also find that models trained with CLM can be transferred to MLM. When given a predetermined amount of computation, and one wants to obtain both a CLM and a MLM model, there is a trade-off in allocating the training token to each model to jointly optimize the performance of the two. Interestingly, the allocation for CLM pre-training was determined by the scaling law of CLM and MLM, and the Effectively Transferred Tokens $D_t$ from CLM to MLM. Furthermore, we verify this method experimentally using a 470M model and fine-tuning on downstream tasks.

- Building on our scaling strategies, we re-allocate of model size and training tokens under the compute budgets of established PROGEN2-xlarge and ESM-2 (3B) setups. Consequently, with the same compute budgets, we trained two corresponding models, one with 7.2B parameters and the other with 10.7B parameters, which exhibited enhanced performance in a diverse range of downstream tasks.

## 2 Scaling up data

First, we explore the effects of training PLMs across multiple epochs under token scarcity conditions. We then introduce a dataset, UniMeta200B, used throughout this work. This dataset enhancement alleviates the challenge of insufficient training for protein language models.

### 2.1 A Data-hungry Observation

Using the UniParc database with 250 million protein sequences, research on ESM [68] shows that the datasets UR50/S and UR50/D, with 45M and 65M unique sequences respectively, outperform Uniref100 in perplexity (PPL) on a ~670M parameter MLM model. These datasets contain ~15B and ~20B unique amino acid tokens. The ESM-2 family models, ranging from 150M to 15B parameters, are trained extensively with nearly 1 trillion tokens over 45 epochs on the UR50/D dataset. In observing the scaling of ESM-2 models, it becomes apparent that increasing model size to 15B parameters from 3B shows marginal improvement. On the other hand, contemporary LLMs are often trained for only one or a few epochs [43, 36, 79, 80, 11]. The repetition of data with limited unique tokens has diminishing returns and hinders scaling model size [65, 34, 58, 70]. This underscores the importance of using rich datasets for training large-scale language models to ensure robust performance across applications. We evaluated models with 150M and 3B parameters on the UR50/S dataset, trained on 200B tokens, as shown in Figure 1. We focus on the Independent and Identically

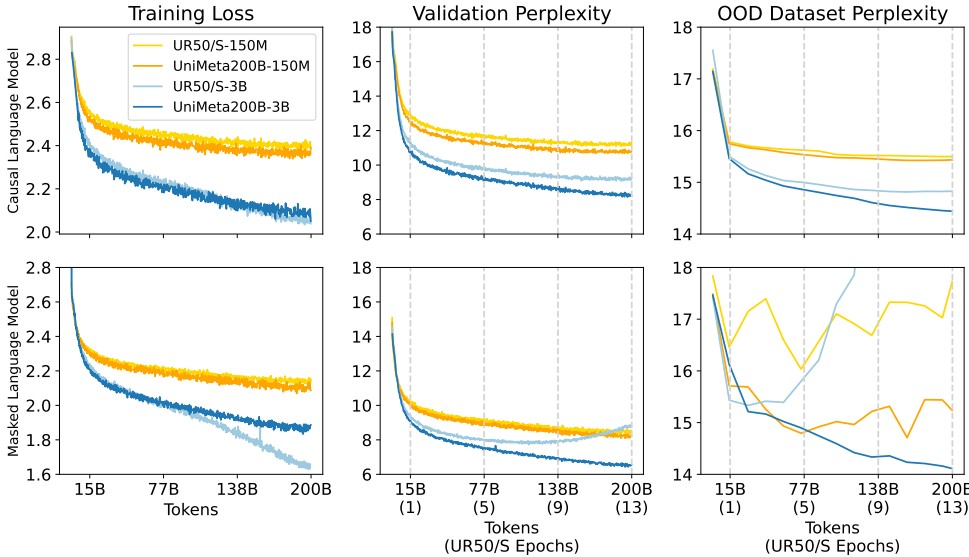

Figure 1: **Learning curves for UR50/S and UniMeta200B.** Training loss and validation PPL, OOD test PPL, were tracked over 200 billion training tokens for both the 150M and 3B models. As we scaled the model from 150M to 3B, we observed diminishing returns on CLM (First line) and a tendency to overfit on MLM (Second line) when repeating the Uniref50 (UR50/S) dataset. We totally evaluate 3 repeating methods on MLM 3B models, all of which present overfitting (see Appendix B).

Distributed (IID) validation and Out-Of-Distribution (OOD) test PPL, which measures the model's randomness in amino acid selection. For our OOD dataset, we utilized the MMseqs2 tool [72] to conduct searches within the UniRef90 database for sequences post-training dataset timestamp, retaining those with *no detectable* identity. From these, a random sample of 3,000 sequences was selected to constitute the OOD dataset. Notably, we do not adopt dropout regularization, a practice that often reduces model capacity and is infrequently used in contemporary LLMs [43]. This choice is consistent with recent LLM configuration findings [38], including ESM-2 [47].

The results show the 150M model lacks good generalization while increasing to a 3B model resulted in diminishing returns for CLM and severe overfitting for MLM. Principally, the bidirectional self-attention mechanisms used in MLM have a higher capacity to overfit compared to the unidirectional self-attention used in CLM. This is because MLM can utilize the entire context surrounding a masked token, leading to faster memorization of the training data.

## 2.2 Expanding Diversified Metagenomic Data

To tackle the challenge of data scarcity, we leveraged the Colab-FoldDB database [56], which focuses on metagenomic data sources such as BFD [1], MGnify [57], and specific eukaryotic and viral datasets including SMAG [22], MetaEuk [44], TOPAZ [3], MGV [59], and GPD [13]. We applied a stringent deduplication process with a maximum similarity threshold of 0.3 to preserve the diversity of the protein universe. Given that the Uniref90 dataset has proven most

Table 1: **The Pre-training data**, aggregates various public sources and specifies sampling proportions for a single epoch of training on 194 billion unique amino acids.

| Datasets | Prot. Seq. | Tokens (AAs) | Samp. Prop. |
|---|---|---|---|
| Uniref50/S | 54M | 15.2B | 8.5% |
| Uniref90/50 | 102M | 37.8B | 19.5% |
| ColabFoldDB$_c$ | 208M | 37.7B | 19.5% |
| ColabFoldDB$_m$ | 575M | 103B | 52.5% |
| Total | 939M | 194B | - |

effective for pre-training across various Uniref clustering levels per ESM-1v [54], we incorporated Uniref90/50 (Before 2022-12), which includes incremental data relative to Uniref50/S representatives. ColabFoldDB$_c$ and ColabFoldDB$_m$ play dominant roles within the dataset, corresponding to cluster representatives and members, respectively. To ensure uniformity during training, we allocate weights within each batch to allow each amino acid token to be evenly processed through the model. This dataset, termed UniMeta200B, contains 939 million unique protein sequences and 194 billion amino acids, which is an order of magnitude larger than UR50/D. We observed significant improvements

in the OOD test set and a consistent learning curve on the IID validation subset extracted from the training set (Figure 1). These enhancements not only ensure a controlled diversity to maintain sample efficiency but also significantly increase the quantity and uniformity of data, facilitating model scaling. [†]

> **Findings 1.** *Scaling the model from 150M to 3B, we noted diminishing returns for CLM and an overfitting tendency for MLM when repeating the UR50/S dataset. The proposed Expanding Diversified Metagenomic Data (UniMeta200B) addresses these problems.*

## 3 Parameters and Datasize Optimal Allocation

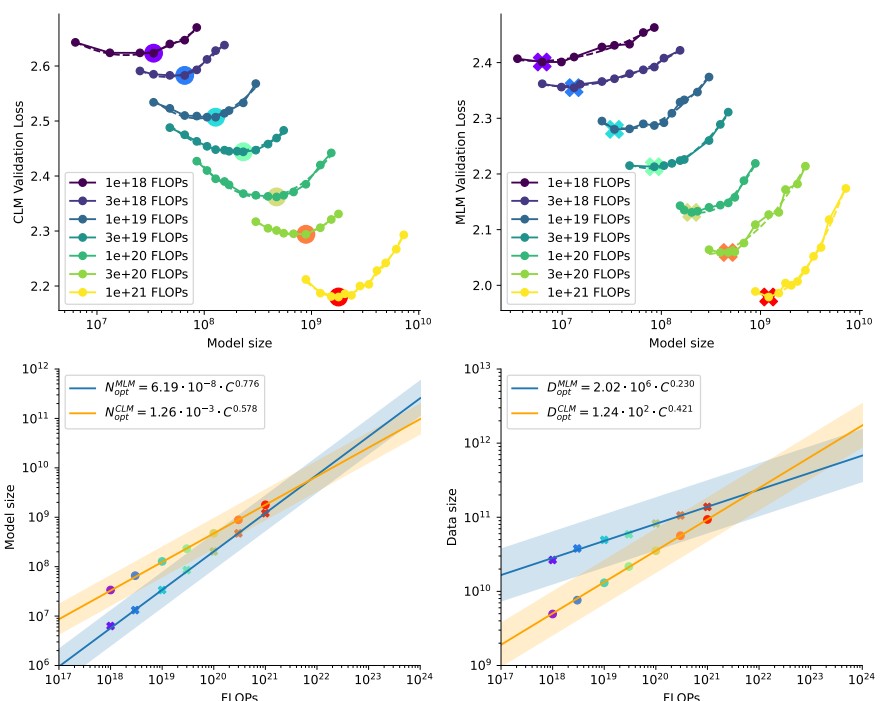

Figure 2: **IsoFLOPs curves and parametric fit for CLM and MLM.** We selected training tokens to ensure a uniform final FLOP count for different model sizes. The lowest loss of each curve revealed an optimal model size for a FLOP budget (**above**). We use these rainbow points at the valley to plot the efficient frontier for estimating the optimal model size and training tokens for scaling models (**below**). The interval range was estimated by model points with similar loss.

In this section, we propose a scaling law for protein sequences with MLM and CLM objectives, aiming at optimally balancing model size and data size under a fixed compute budget to improve efficiency on expanded resources.

### 3.1 Scaling laws for CLM and MLM

We first fit our models in the form of a fundamental power-law based on the existing work [42, 36, 33, 69, 2, 17, 90] in the field of LLMs. Specifically, given a fixed FLOPs formula of $C = 6 \times N \times D$, where $N$ represents the

Table 2: Coefficient of Equation 1.

| Parameter | $\alpha$ | $\beta$ | $A$ | $B$ |
|---|---|---|---|---|
| CLM | 0.578 | 0.422 | $1.26 \times 10^{-3}$ | $1.23 \times 10^{2}$ |
| MLM | 0.776 | 0.230 | $6.19 \times 10^{-8}$ | $2.02 \times 10^{6}$ |

number of forward-activated non-embedding parameters, and $D$ is the number of training tokens, how should one navigate the trade-off between model size and the number of training tokens? The

---

[†]Appendix E compare the training performed separately on two datasets, and we find that the ColabFoldDB does not affect downstream results.

model parameters $N$ and data size $D$ can be directly fit with a simple power-law:

$$N(C) = A \times C^{\alpha}, \quad D(C) = B \times C^{\beta} \tag{1}$$

We employed the IsoFLOPs profiling approach [36, 9], setting 7 distinct training FLOP counts ranging from $1 \times 10^{18}$ to $1 \times 10^{21}$. For each FLOP count, we selected models from a pool of candidates (see Appendix N). Models were excluded if the estimated data size $(C/(6*N))$ resulted in more than 200B tokens or if the training steps were fewer than 20K. Ultimately, approximately 260 models were used for fitting. We considered the final validation loss for each model to ensure that every model completed a full cosine cycle with $10\times$ learning rate decay. For each fixed FLOP count, we employ smoothed loss to determine the optimal model size with the smallest loss (Figure 2 (above)). Subsequently, we use Equation 1 and apply the `least_squares` method to fit the model. Given

Table 3: Coefficient of Equation 2

| Objective | $\alpha_N$ | $\alpha_D$ | $\alpha_C$ | $\beta_N$ | $\beta_D$ | $\beta_C$ |
|---|---|---|---|---|---|---|
| CLM | $-0.037$ | $-0.051$ | $-0.027$ | $4.835$ | $7.904$ | $8.251$ |
| MLM | $-0.040$ | $-0.120$ | $-0.034$ | $4.530$ | $42.614$ | $10.125$ |

the minimal variations in the final loss among a set of $(N, D)$ configurations, we classify these configurations as operating under "IsoLoss" conditions (see Appendix K Figure A15), considered optimal for training. In Figure 2 (below), we illustrate an efficient frontier interval that demonstrates permissible fluctuations in model size and dataset size at a specific FLOP count, while still achieving nearly identical losses. The variation in loss is quantified at 0.25 on a logarithmic scale with a base of 10. This indicates that within this FLOP counts, the model size can be adjusted within a range, increasing up to 80% or decreasing up to 40% without repeating data, to maintain a loss variation within 0.01.

We observe distinct growth rates in the proportional relationship between model size and training tokens for the MLM model compared to the CLM, as detailed in Table 2. Both models demonstrate an increase in the growth of model size that surpasses the growth of training tokens. Up to the intersection point around $1 \times 10^{22}$ (see Figure 2, left below), the model size of MLM tends to be smaller than the CLM, thereafter, the MLM rapidly exceeds that of the CLM. Notably, the growth of the MLM's training tokens is greatly lower than that for the CLM, possibly due to MLM's higher sample efficiency. For instance, if the compute budget is increased by $10\times$, the size of the CLM model should increase by $4\times$

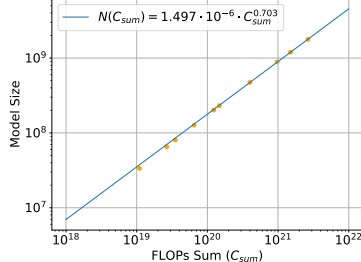

Figure 3: Compute allocation for two objectives with the same model size.

and the training data by $3\times$, aligning more closely with equally proportional scaling. For the MLM, the model size should increase by $6\times$ and the training data size by $1.7\times$.

In exploring the scaling relations of loss, we analyzed various model sizes $N$, compute budgets $C$, and training dataset tokens $D$. These can be described by a similar power-law relation defined as:

$$L(x) = \beta_x \times x^{\alpha_x} \tag{2}$$

where $\alpha_x$ is the scaling exponent for different variables. For each FLOP count, we aimed to identify the minimal loss as the fitting target along with the corresponding independent variable $x$. Table 3 presents these fitting coefficients.

Based on the coefficients obtained from the fitting described above, we can establish the relationship between $D$ and $N$ by eliminating $L$. The relationship is expressed by the following equation:

$$D(N) = \left(\frac{\beta_N}{\beta_D}\right)^{\frac{1}{\alpha_D}} \times N^{\frac{\alpha_N}{\alpha_D}} \tag{3}$$

By substituting the learned coefficients into this formula, we can derive $D_{\text{MLM}}^{\text{opt}}$ and $D_{\text{CLM}}^{\text{opt}}$ when given $N$. The estimation may be affected when the data exceeds 200 billion or when the quality or quantity of the training dataset changes.

## 3.2 Scaling law for training two models

When our goal is to optimize both CLM and MLM simultaneously, the strategic allocation of compute resources between these two objectives becomes essential. To facilitate this, we equalize model parameters across objectives to assess specific compute budgets for dual-objective training. Specifically, we seek the compute budgets, $C_{\text{MLM}}$ and $C_{\text{CLM}}$, for configurations where the optimal model size is the same, i.e., $N(C_{\text{MLM}}) = N(C_{\text{CLM}})$. These individual computations are then aggregated to formulate the overall compute budget:

$$C_{\text{sum}}(N) = C_{\text{MLM}}(N) + C_{\text{CLM}}(N) = \left(\frac{6.2 \times 10^{-8}}{N}\right)^{0.776} + \left(\frac{1.25 \times 10^{-3}}{N}\right)^{0.578} \tag{4}$$

These two objectives share the same parameter size, their compute budget $C$ and the number of training tokens $D$ differ. Thus we further introduce a model-to-ratio $r(N)$ as $D_{\text{MLM}}(N)/D_{\text{CLM}}(N)$. We then achieve the relationship between $N$ and $C_{\text{sum}}$ by a fitted power-law (Figure 3) form:

$$\begin{cases} N(C_{\text{sum}}) \approx 1.497 \times 10^{-6} \times C_{\text{sum}}^{0.703} \\ r(N) \approx 8.449 \times 10^{4} \times N^{-0.392} \end{cases} \tag{5}$$

The ratio $r(N)$ informs us about the allocation proportion of training tokens. Specifically, under equal parameters, the data for MLM should exceed that for CLM until a 10B threshold (achieving a 1:1) is reached, after which more training tokens are allocated to CLM.

We further find that the scaling behavior of sparse parameter counts in Mixture of Experts (MoE) protein model [74], configured with eight experts (see Appendix I), as well as a combined power-law formula used to fit our data (see Appendix J), both exhibit a similarity to the scaling behavior we have proposed.

> **Findings 2.** *In both CLM and MLM, training data scales sublinearly with model size, following distinct power laws. With an "infinite" dataset, where samples are not repeated and training for less one epoch, MLM's model size grows faster than CLM's.*

## 4 Transfer Scaling

We have outlined two independent scaling laws and how to allocate FLOPs under a fixed budget for training two optimal models, one with MLM and the other with CLM. However, we have not explored the interaction between these objectives. This raises important questions: Can models trained with one objective transferred to one with another objective? Is there a synergistic effect from training two models? Does training order impact the results?

### 4.1 Transferability

We conduct transfer learning experiments on MLM and CLM objectives, selecting eight optimal model sizes based on Equation 1. These models correspond to four increasing FLOP counts from $3 \times 10^{19}$ to $1 \times 10^{21}$ and undergo training from scratch followed by transfer training. Transfer training involves initially training on MLM or CLM, then training on the alternate model for each size.

We find that optimal pre-training on one objective benefits the target objective in transfer learning, though effects vary between methods. Starting with CLM and then training MLM, benefits increase with model scale. In contrast, starting with MLM then training CLM sees diminishing benefits. As shown in Figure 4 (left), for a model size of 230M with $3 \times 10^{19}$ FLOPs, CLM from MLM pre-training reduces the loss by 0.02 compared to CLM from scratch, however, benefit that nears zero for the 1.7B model. Conversely, for models from 85M to 1.2B, transfer benefits grow with model size, the compared validation loss gap increasing from 0.025 to 0.045. This likely stems from the higher loss utilization rate in CLM; CLM calculates losses for all tokens in a protein sequence, whereas MLM only calculates losses for 15% of the tokens. [‡].

---

[‡]Appendix C analyzes the mask ratios.

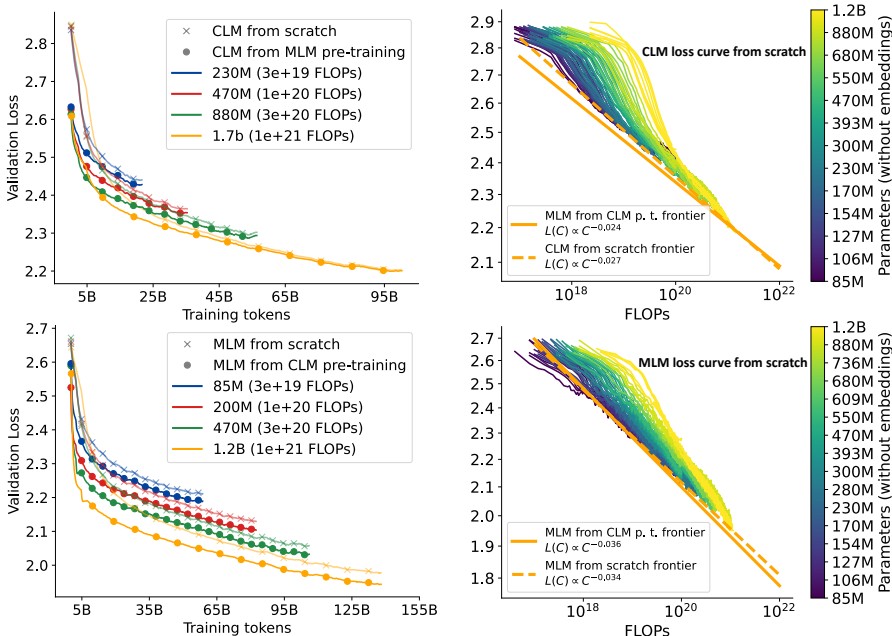

Figure 4: **Left:** The upper graph compares validation loss of CLM trained from scratch with those transferred from MLM, showing diminishing transfer benefits as model size increases. The lower graph depicts increased benefits for MLM from pre-trained CLM with larger sizes, indicating scale-dependent efficiency gains. **Right:** Shows loss curves for CLM and MLM across different FLOPs, emphasizing the efficient frontiers (or Pareto Frontier) from various transfer strategies. It highlights that the benefits of transferring from CLM to MLM grow with model size, reflecting a scale-dependent synergy between training objectives.

We use a power-law to model the transfer scaling law, initially excluding the pre-training FLOPs. The scaling behavior of transfer learning is modeled by:

$$L(C_s) = A_s \times C_s^{\alpha_s}, \quad L(C_t) = B_t \times C_t^{\alpha_t} \tag{6}$$

where $L(C_t)$ and $L(C_s)$ represent the loss for transfer learning and training from scratch.

Figure 4 (right) shows that the efficient frontier for $L(C_t)$ has shifted relative to $L(C_s)$ (it can be directly obtained from Table 3, repeated here for convenience.), indicating an improvement. The coefficients from both are shown in Table 4,

Table 4: Coefficients for $L(C_s)$ and $L(C_t)$

| Parameter | $A_s$ | $\alpha_s$ | $B_t$ | $\alpha_t$ |
|---|---|---|---|---|
| MLM | 10.125 | $-0.034$ | 11.133 | $-0.038$ |
| CLM | 8.251 | $-0.027$ | 7.191 | $-0.024$ |

where we can infer that $C_t \propto C_s^{\frac{\alpha_s}{\alpha_t}} = C_s^{0.89}$, suggesting that training MLM from scratch with $10\times$ the compute requires approximately $7.7\times$ the compute compared to MLM from CLM pre-training. Another observation is that mixing training objectives in a single batch tends to be detrimental. Detailed results and settings are in Appendix H. The recommended transfer learning schedule involves pre-training CLM before MLM, as mixed training and order swapping show no benefits. We speculate that this primarily occurs because our MLM, which focuses solely on recovering corruption tokens, is not causal. If it predicted a middle segment in a left-to-right manner, it could mutually adapt with the context to accelerate training [86].

> **Findings 3.** *Transferring from MLM to CLM results in diminishing returns. Conversely, transferring from CLM models to MLM models remains effective as compute scales.*

## 4.2 Effectively Transferred Tokens

Although we observe that MLM benefits from transfer learning from CLM, the pre-training compute budget remains unaccounted for. We focus on two aspects: (1) the actual benefit CLM provides to MLM and its predictability, and (2) performance differences between MLM trained from pre-trained

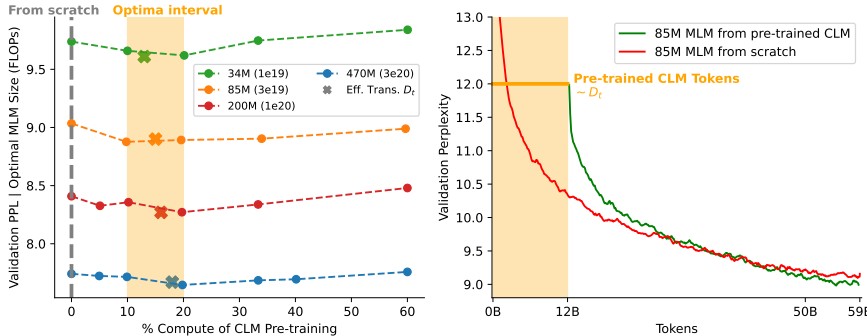

Figure 5: **Left**: Valid perplexity of % compute allocated for the CLM pre-training. For instance, % compute indicates first training on CLM and then the rest compute fine-tuning on MLM. The optimal CLM pre-training % compute range with [10, 20]. And the fitted $D_t/(D_t + D_f)$ drops in the optimal loss range. **Right**: Comparison of validation perplexity for models trained from scratch (red) and those fine-tuned from a pre-trained CLM (green), demonstrating that fine-tuning from a CLM reduces perplexity with similar or even fewer tokens.

CLM (MLM-CLM) and MLM from scratch (MLM-S) under identical FLOP constraints. We define Effectively Transferred Tokens $D_t$ as the *additional data a model of the same size would need to train from scratch on MLM to achieve the same loss as a model pre-trained on CLM.* If the token number in the pre-trained CLM model exceeds $D_t$, then the computations for CLM pre-training was excessive. Knowing $D_t$ in advance would guide the allocation of tokens for CLM pre-training.

We compare MLM-S and MLM-CLM models ranging from 33M to 1.2B with FLOP counts from $3 \times 10^{19}$ to $1 \times 10^{21}$. By calculating the *token distance* at the same loss level between these models, we establish our fitting target $D_t$, collecting approximately 2800 sample points. Following similar methods in scaling transfer works [35, 90], $D_t$ is defined by a simple multiplicative scaling formula:

$$D_t = k \times \frac{1}{D_f^\delta} \times \frac{1}{N^\gamma}; \quad k \approx 3.65 \times 10^5, \quad \delta \approx -0.137, \quad \gamma \approx -0.369 \qquad (7)$$

where $D_f$ represents the tokens used for MLM-CLM, and $N$ is the number of parameters, with $k$, $\delta$, and $\gamma$ as fitting coefficients. For instance, a $10\times$ increase in $D_f$ would roughly triple the model size and double $D_t$. We validate these findings by evaluating the compute ratio of CLM pre-training under four specified parameters and FLOPs, as shown in Figure 5 (left), finding that MLM-CLM generally outperforms MLM-S. Specifically, $D_t/(D_t + D_f)$ ranges from 10% to 20% of the compute budget for CLM pre-training. Figure 5 (right) schematically illustrates the learning curves of two 85M (3e19 FLOPs) models, with MLM-CLM achieving similar or better loss levels with equal or fewer tokens.

---

**Findings 4.** *Training MLM from scratch with 10× the compute requires approximately 7.7× the compute compared to MLM from CLM pre-training, implying that around 20% of the compute budget should be allocated for CLM pre-training to get better MLM models transferred from CLM pre-training.*

---

## 5 Experimental Validation

Based on the scaling laws we observe, we estimate the model size and training tokens for current leading models by analyzing their FLOPs. In our configuration, the PROGEN2-xlarge model, with 6.4B parameters, is estimated to require training with 7.2B parameters and 265B tokens. Similarly, the ESM-2 model, with 3B parameters, should be trained with a model size of 10.7B parameters and 260B tokens. Additionally, we employed two 470M models to test the transfer scaling strategy, one trained from scratch (470M scratch) and the other from CLM pre-training (470M trans.). The model's details are reported in Table 5.

Table 5: **Model architecture details.** We compare popular models PROGEN2 and ESM-2 using similar FLOPs with our models estimated by proposed scaling law.

| Params | Objective | $N_{\text{head}}$ | Dim. | $N_{\text{layer}}$ | Train. Tokens | FLOPs |
|---|---|---|---|---|---|---|
| PROGEN2-xlarge (6.4B) | CLM | 16 | 4096 | 32 | 350B | $1.34 \times 10^{22}$ |
| Our 7.2B | CLM | 32 | 4096 | 36 | 265B | $1.14 \times 10^{22}$ |
| ESM-2 (3B) | MLM | 40 | 2560 | 36 | 1T | $1.68 \times 10^{22}$ |
| Our 10.7B | MLM | 32 | 4352 | 47 | 260B | $1.68 \times 10^{22}$ |
| 470M scratch | MLM | 16 | 1280 | 24 | 106B | $3.0 \times 10^{20}$ |
| 470M Trans. | CLM + MLM | 16 | 1280 | 24 | 21B + 85B | $3.0 \times 10^{20}$ |

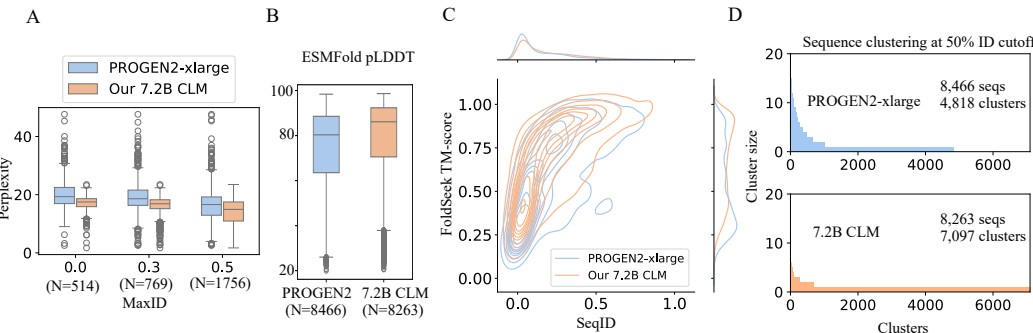

Figure 6: **Comparative Analysis of CLM Models. A.** Perplexity analysis for PROGEN2-xlarge and our 7.2B CLM shows lower values for our model across various MaxID levels, suggesting better sequence handling. **B.** Box plots of pLDDT scores for protein structures by PROGEN2-xlarge and our 7.2B CLM. **C.** Contour and line plots show our 7.2B CLM sequences mimic natural sequences more closely than PROGEN2-xlarge, assessed using Foldseek with the PDB database. **D.** Clustering at 50% sequence identity reveals our 7.2B CLM generates more clusters, indicating higher diversity.

## 5.1 Protein Generation Comparison: 7.2B CLM vs. 6.4B PROGEN-xlarge

We first evaluate the perplexity on OOD data and then compare the protein generation capabilities of the 7.2B CLM and PROGEN2-xlarge models. Each model generated 2,000 sequences for each parameter combination of top-$p$ $\{0.5, 0.7, 0.9, 1.0\}$ and temperature $t$ $\{0.2, 0.4, 0.6, 0.8, 1.0\}$, totaling 40,000 sequences per model. Sequences with a perplexity greater than 10 and duplicates were removed, leaving 8,263 and 8,466 sequences for the 7.2B CLM and PROGEN-xlarge, respectively. We used four metrics to assess the quality of the models and the generated sequences (See Appendix F for details).

**OOD Dataset PPL Analysis** We randomly sampled 5,000 sequences from UniProt released after 2023-01-01 and aligned them to our and PROGEN2's training data (Uniref90 and BFD) using HHblits [66] or Jackhmmer [29]. Sequences below a maximum identity cutoff were used to assess the models' PPL, as shown in Figure 6A. Our 7.2B CLM exhibited lower PPL on three subsets.

**pLDDT scores from ESMFold** Atomic structures of 8,263 and 8,466 generated sequences were predicted using ESMFold, and compared based on pLDDT scores, displayed in Figure 6B. The 7.2B model's average pLDDT score was 78.69, higher than PROGEN2-xlarge's 74.33.

**Natural Sequences Comparisons with Foldseek** Using Foldseek [81], we searched the PDB database for sequences similar to those generated by our 7.2B CLM model, which showed better mimicry of natural sequence properties with higher average TM-scores (0.655 vs 0.522) and SeqID (0.194 vs 0.165), as shown in Figure 6C.

**Diversity Analysis** Generated sequences were clustered using MMseqs2 [72] with a 50% similarity cutoff. The 7.2B CLM model resulted in higher diversity with 7,097 clusters compared to 4,818 clusters for PROGEN2-xlarge, detailed in Figure 6D.

## 5.2 Protein understanding tasks: 10.7B MLM vs. 3B ESM2

We evaluate different task types from the protein benchmark [14]: Contact prediction as binary classification at the amino acid pair level; fold classification into 1195 classes at the sequence level; and fluorescence as regression tasks. Following [14], we add a Multi-Layer Perceptron (MLP) head to each pre-trained model and apply Low-Rank Adaptation (LoRA) [37] (r=8, $\alpha$=16) for fine-tuning (see Appendix G for convergence details).

Table 6: Tasks performance of MLM Model on the test dataset with LoRA fine-tuning.

| Models | Contact Pred. (P@L/5) | Fold Class. (1195 class.) | Fluor. (reg.) |
|---|---|---|---|
| ESM-2 (3B) | 0.91 | 0.69 | 0.65 |
| Our 10.7B | 0.91 | **0.72** | **0.69** |
| 470M scratch | 0.78 | 0.65 | 0.67 |
| 470M trans. | **0.80** | **0.66** | 0.67 |

The results, shown in Table 6 and A7, demonstrate that our 10.7B model outperforms ESM-3B on 7 out of 8 tasks. This confirms the rationale behind the observed scaling law and addresses concerns about the scope and rigor of our evaluation tasks. Additionally, the 470M model transferred from CLM pre-training continues to perform effectively in this task, showing the efficacy of the observed transfer scaling law.

## 6 Discussion and Limitations

**Data Repeat Scaling Law.** Our scaling law is derived within a single-epoch training setting. It is well known that MLM exhibits higher sample efficiency than CLM due to its dynamic masking strategies across multiple epochs. However, this advantage diminishes when training is restricted to just one epoch. We present an empirical study comparing a 2.8B model trained on 1T tokens (approximately five epochs) with a 10.7B model trained on 265B tokens (roughly 1.4 epochs). The two models achieve similar performance in terms of OOD PPL (10.33 vs. 10.21) while utilizing the same amount of FLOPs. This finding suggests that repeating multiple rounds of MLM training has minimal impact on reducing loss. Notably, the smaller models are more user-friendly during inference and fine-tuning. Therefore, we suggest an alternative approach that adjusts the optimal training token count and model size within the data-repeat scaling law.

**Multi-modality Scaling.** The multi-modal auto-regressive work [33] suggests the existence of a nearly universal scaling law across various modalities, including images, videos, math, code, and languages. Our results appear in this trend as well, such as, the scaling laws for CLM exhibit similarities to those in natural languages. The same situation may apply to other modalities of biological data, such as RNA and DNA [60].

**Various Pre-train Datasets and Strategies.** Our datasets cover a substantial portion of the protein universe, yet they might not be entirely representative. Combining BFD [8], Uniref [75], MetaClust [44], and IMG/JGI [52] with 90% clustering results in at least 600 billion unique tokens. However, variations in datasets could affect the power-law behavior. Future work could explore applying our findings to different model architectures. There is ongoing research on scaling LLMs for long sequences [7, 15, 16, 18, 39, 48, 87], and MSA augmentation could significantly improve protein representation regarding contacts and structure. Investigating scaling laws in this context could be a promising direction for future research.

## 7 Conclusion

In this work, we are the first to establish a practical pathway for researchers to develop faithful and powerful protein language models optimized by both CLM and MLM objective in an end-to-end manner. This includes everything from pre-training dataset construction, expanded metagenomic databases such as ColabFoldDB, emphasizing the critical importance of data quality and quantity for scaling language models, to optimal parameter and dataset allocation along with the potential loss prediction, as well as knowledge transfer from other pre-training objectives. Our work holds significant potential for the application of large language models across various scientific domains.

**Acknowledgments.** This work has been supported by the National Key R&D Program of China 2021ZD0113304, NSFC for Distinguished Young Scholar 62425601, New Cornerstone Science Foundation through the XPLORER PRIZE.

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

Table A7: Tasks performance of MLM Model on the 5 test dataset, Fitness Prediction (Fit P.) as a regression task measured by Spearman coefficient at sequence level, Localization (Loc.) as 10 sub-cellular classification task at sequence level, Metal Ion Binding (MIB) as a binary classification task at sequence level, Solubility (Sol.) as a binary classification task at sequence level, Stability (Sta.) as a regression task measured by Spearman coefficient at sequence level, with LoRA fine-tuning.

| Model | Fit P. (SP) | Loc. (ACC) | MIB (ACC) | Sol. (ACC) | Sta. (SP) |
|---|---|---|---|---|---|
| ESM2 (3b) | 0.94 | **0.81** | 0.82 | 0.74 | 0.82 |
| Our 10.7B | **0.96** | 0.79 | **0.83** | **0.79** | **0.83** |

# Appendix

## Table of Contents

## A    Related Work

**Protein Language Model** Since the advent of AlphaFold2 [41], the masked language model (MLM) has been integrated as a subtask within the Evoformer architecture. In this context, an assumption is that large language models can be considered as a lossless compression method [21]. This was followed by a series of language modeling efforts [31, 10, 32, 28, 27], which aimed to conduct pre-training on single-sequence proteins using larger datasets and model scales. These efforts sought to harness the scale of the models to learn complex co-evolutionary information, although detailed investigations on how to optimally scale these models remain scarce. Our work primarily focuses on these finer aspects, aiming to fill this gap in the research.

**Training objectives** In natural language processing (NLP), masked language models (MLM) are rarely adopted due to the self-explanatory nature of natural language, which inherently prompts the meta-knowledge of tasks and generates task targets through CLM (Conditional Language Modeling) training models. However, a unified language modeling objective for Protein Language Models has yet to be fully consented. Those based on causal language modeling (CLM) have been primarily explored for protein design. Benchmarks in protein design using MLM [84] have also shown promising results for generation [62], exhibiting variable performance when compared to CLM [91, 83]. Additionally, the potential of the in-filling task objective remains largely unexplored [6, 77, 25]. Our research aims to thoroughly discern the scaling behavior of the two most common optimization objectives in this domain.

**Scaling Laws** To our knowledge, the concept of scaling laws of language model is first introduced by OpenAI [42]. Subsequently, numerous variants and modifications [36] have been developed around this theme. Recently, an array of new scaling laws has emerged. These include scaling laws related to learning rates and batch sizes [9], data-constrained scaling laws [58], scaling laws for downstream tasks and Transfer [90, 35], as well as scaling laws within the Mixture of Experts (MoE) framework [17], and those concerning long sequences and positional encoding [49]. While these laws are primarily derived using auto-regressive models in resource-rich domains, their application in the biological data sector is less common. Our work seeks to address this gap. Furthermore, scaling laws for Masked Language Models (MLM) are notably scarce. Given that MLMs are currently one of the most effective training methods for biological data, our research on MLMs could also be extended to other non-text domains.

# B    UR50/S Repeat Experiments

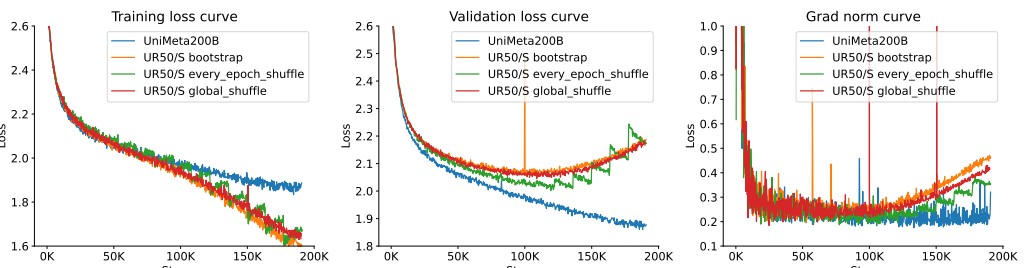

Figure A7: **Learning curve for UR50/S dataset repetition methods**. Our 194B tokens dataset (UniMeta200B) shown in blue, serves as the reference with an approximate single epoch run. The bootstrapping method, depicted in orange, processes 200 billion tokens with replacement, indicating a tendency towards zero unsampled by the fifth epoch. The every-epoch shuffle method, in green, ensures all tokens are used per epoch, forming a stair-step pattern in training loss. Lastly, the global shuffle method, in red, loosely uses all tokens each epoch but ensures the strict number of epoch passes for every token. The rightmost plot of gradient norms shows an uptick for curves corresponding to overfitting, signifying a lack of further optimization, with steep or erratic gradients indicated by the ascending gradient norms.

We employed three different methods to repeat training on the UR50/S dataset, all of which ultimately led to overfitting. The reference for these experiments is shown by the blue curve in Figure A7, which represents UniMeta's loss for approximately one epoch.

Firstly, using bootstrapping, we processed 200 billion tokens from UR50/S with replacement. In each epoch, 65% of the dataset was randomly selected, leading to a diminished proportion of unsampled tokens by the fifth epoch, as depicted by the orange curve.

Secondly, we shuffled the unique data for each epoch to ensure that all UR50/S tokens were used per epoch, resulting in a stair-step pattern [30] in the training loss, illustrated by the green curve. It has simply memorized the dataset but isn't improving at generalizing. Over-confident predictions of the first batch of the next epoch lead to a big step update, and then the model is not adapted to the next batches, resulting in no longer a decrease in loss.

Lastly, we shuffled the entire training dataset less stringently, which did not strictly ensure that all UR50/S tokens were used every epoch, but guaranteed that each token was used an equal number of times over the entire training period. We term it global shuffle, this approach is shown by the red curve.

From the gradient norm curve shown in Figure A7 (right), we observe an uptick in gradient norm for the overfitting curves, indicating that the model is no longer optimizing effectively. In machine learning, such an increase in gradient norm typically suggests that the model is encountering areas of the parameter space where gradients are steeper or more erratic, often occurring when the model starts to memorize the training data rather than generalize from it, approaching a saturated network [55]. This behavior can result from overly complex models, too many training epochs without sufficient regularization, or training on non-representative data.

## C   Choice of Masking Ratio

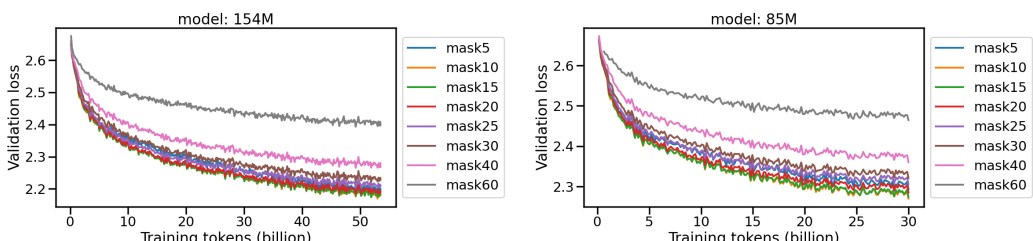

Figure A8: **Validation loss of different masking ratios**. Two models (154M and 85M) are trained from 5% to 60% masking intervals.

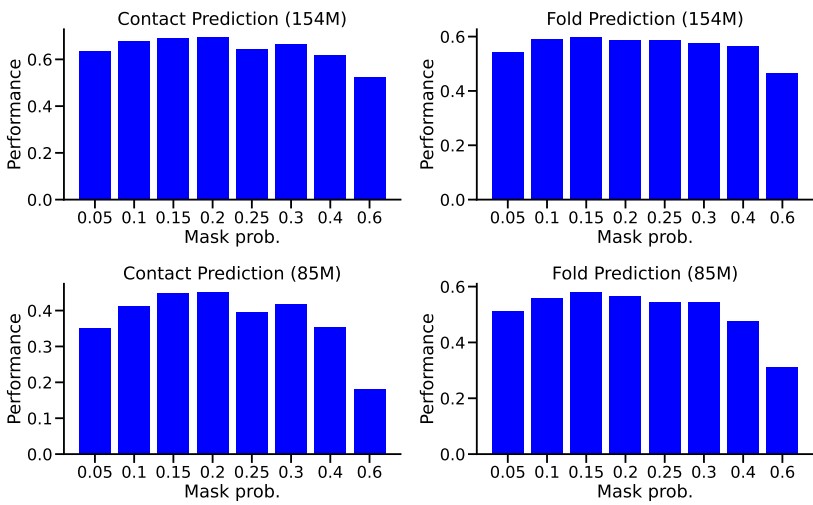

Figure A9: **Abalation of different masking ratios**. Two models (154M and 85M) are trained from 5% to 60% masking intervals, and evaluated on contact map and fold classification downstream tasks.

In the original BERT work [23], the absence of masked tokens in downstream tasks presented a mismatch with the pre-training data distribution. The authors investigated various masking ratios and concluded that a 15% masking rate was most beneficial for downstream tasks. This was implemented alongside an 80-10-10 strategy: 80% of the tokens were replaced with a mask, 10% were randomly substituted, and the remaining 10% were left unchanged.

However, given the significant differences between protein sequences and natural language processing data, we employed two models, sized at 85M and 154M, to explore a range of masking ratios from

5% to 60% (see Figure A8). The best masking ratios for validation loss drop ranged from 10% to 20%; ratios too small (5%) or too large (greater than 25%) degraded the performance.

We further used pre-trained eight different models to perform full fine-tuning on downstream tasks such as Contact Prediction and Fold Classification in Figure A9. Results from the test datasets revealed that, similar to NLP, the optimal performance was achieved within a 10%-20% masking range. Specifically, a 20% masking ratio slightly outperformed 15% in Contact Prediction, while the 15% ratio yielded the best results in Fold Prediction. Consequently, for our Masked Language Model (MLM), we decided to adhere to the 15% masking ratio with the 80-10-10 strategy for training all our models.

## D    MLM/CLM for Protein Contact Prediction

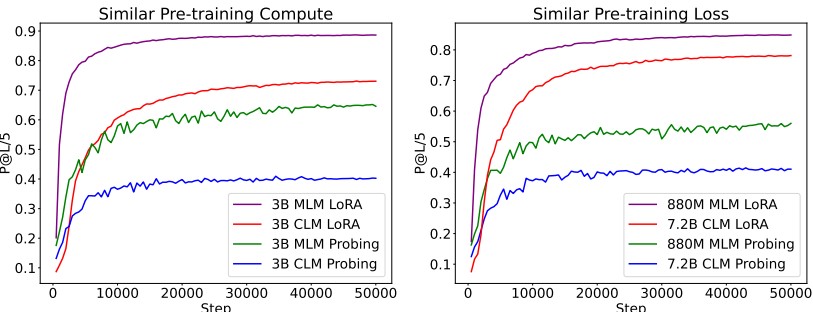

Figure A10: **Contact Prediction on MLM and CLM models.** Two 3B models (CLM and MLM) were trained using identical computational resources, represented by the probing and LoRA fine-tuning methods. On the right, performance of a 7.2B CLM model is compared with an 880M MLM model under similar pre-training loss conditions. These models exhibit differing rates of convergence, highlighting the impact of uni-directional and bi-directional model architectures on learning dynamics.

We compared the effectiveness of CLM in the downstream task of contact prediction, using two different setups (Figure A10). In the first setup, two 3B models were trained under identical computational resources on 200 billion tokens, $3.4 \times 10^{21} FLOPs$. Their performance was evaluated through two training approaches: Probing (freezing the pre-trained model) and LoRA fine-tuning, with an added MLP head for comparison.

In the second setup, we compared the effects of MLM and CLM under similar loss conditions. Here, a 7.2B CLM model and an 880M MLM model were selected, both achieving a loss of 1.98 on our validation set. Despite the MLM model having a simpler loss calculation, involving a 15% mask rather than a one-by-one mask—which would result in a higher loss—the MLM significantly outperformed the CLM. Importantly, the CLM model's computational power was an order of magnitude greater than the MLM model ($1.68 \times 10^{22}$ vs $1.0 \times 10^{21}$ FLOPs). This suggests that despite the lower loss achievable by the CLM model compared to MLM with a one-by-one mask, the unidirectional limitations of CLM do not translate into better downstream task performance.

## E    Pre-training Dataset Quality

Compared to Uniref90, ColabFoldDB offers a higher diversity and larger numbers of protein sequences, though with generally shorter sequence lengths, likely suggesting potentially lower data quality. To evaluate the efficacy of our expanded dataset, ColabFoldDB, we initially trained two 85M models separately on Uniref90 and ColabFoldDB. Uniref90 in our dataset comprises two subsets: Uniref50/S and the incremental dataset over Uniref50/S, termed Uniref90/50. Similarly, ColabFoldDB consists of representative and member data. We controlled the sampling proportion to ensure uniform sampling across both datasets, with results reported in Table A8. Both models were then trained using identical configurations on a 50B scale.

From the perspective of validation loss in pre-training, the higher loss on ColabFoldDB might be attributed to its lower diversity and shorter sequence lengths compared to Uniref90. However, the performance on downstream tasks, such as contact prediction and fold classification, shows negligible differences between models trained solely on ColabFoldDB and those trained on Uniref90, as illustrated in Figure E. This confirms that ColabFoldDB is an effective expansion of Uniref90 that maintains sample efficiency.

Table A8: **Compared two dataset characteristics.** Protein sequence count, token number, and sampling proportions for Uniref50/S, Uniref90/50, and ColabFoldDB representative and member data.

| Datasets | Prot. Seq. | Tokens (AAs) | Sampling Prop. |
|---|---|---|---|
| Uniref50/S | 54M | 15.2B | 28.67% |
| Uniref90/50 | 102M | 37.8B | 71.33% |
| ColabFoldDB$_c$ | 208M | 37.7B | 26.75% |
| ColabFoldDB$_m$ | 575M | 103B | 73.52% |

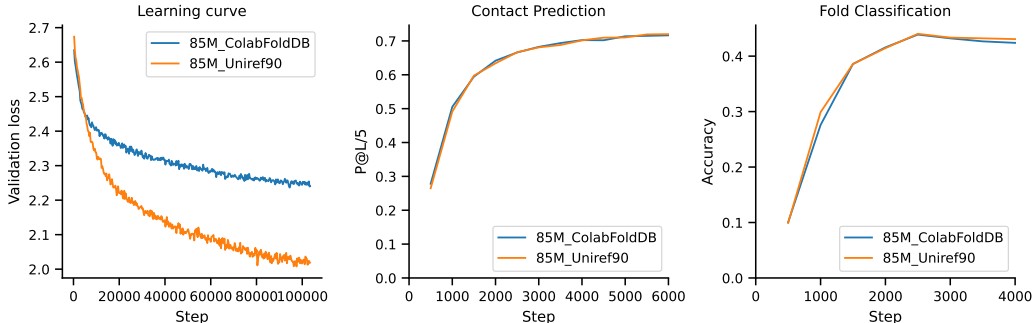

Figure A11: **Data quality check.** Comparison of learning dynamics and downstream task performance for two 85M models trained on ColabFoldDB and Uniref90. Left: Validation loss curves demonstrating initial training differences. Middle: Contact prediction performance showing the response to testing on similar tasks. Right: Fold classification accuracy, comparing model responses to structural prediction tasks. Despite initial differences in loss, both datasets yield comparable performance in downstream applications.

## F  The Evaluation of Protein Generation.

We explain more details about Protein Generation Comparison as follows:

*OOD Dataset PPL Analysis.*  PPL represents the probability of the test sequences in the model distribution. The lower the PPL, the closer the distribution of the model and the test set is. In order to test the generalization ability of the model on new data, we use different sequence identity (0.0, 0.3, 0.5) as thresholds to select the test set.

*pLDDT Scores from ESMFold.*  Predicted Local Distance Difference Test is the confidence level of ESMFold protein structure prediction. This metric is widely used in methods such as AlphaFold2, ESMFold, and OpenFold. pLDDT filters are often used in protein design (such as RFDiffusion), which can significantly improve the success rate of protein design;

*Natural Sequences Comparisons with Foldseek.*  Foldseek takes protein structure as input and searches for proteins with similar structure in the database. We use the experimentally-resolved protein structure as the database (PDB database) to explore how the structure of the generated sequences close to PDB (a higher TM-score indicates higher structural similarity). This method has been used to evaluate other methods for protein sequence generation (ProGen2, ProtGPT2);

*Diversity Analysis.*  We cluster the two sets of sequences (ProGen2-xlarge and CLM) according to sequence similarity. Sequences with a identity higher than 50% will stay in one cluster. Since the

number of input sequences is similar (8,466 vs 8,263), we can measure the diversity of the generated sequences by comparing the number of clusters.

# G    Convergence Analysis of Downstream Fine-tuning Tasks

Observing the learning curves in Figure A12a, we can assess the effectiveness of different fine-tuning scenarios. For the contact prediction task, the convergence speed under the LoRA setting is very similar for both models. Our testing reveals closely matching results for ESM-2 models with capacities of 650M, 3B, 15B, consistent with the findings reported by Ankh et al. [27]. This similarity suggests possible saturation of the dataset under single-sequence pre-trained models. Additionally, the convergence rates for tasks such as fold classification and fluorescence are generally faster than those for ESM-2, indicating robust generalization following our data augmentation strategies.

Based on the two 470M models defined in our Table 5, despite using the same computational power, we observe distinct outcomes (Figure A12b) in contact prediction and fold classification tasks. The MLM model from CLM pre-training converges slightly faster than MLM from scratch. However, the distinction is less pronounced in the two downstream regression tasks. This suggests that perplexity is more sensitive to protein structure related tasks, i.e., contact prediction and fold classification, but shows less sensitivity to regression tasks, particularly when assessed using the Spearman metric, which is prone to variability.

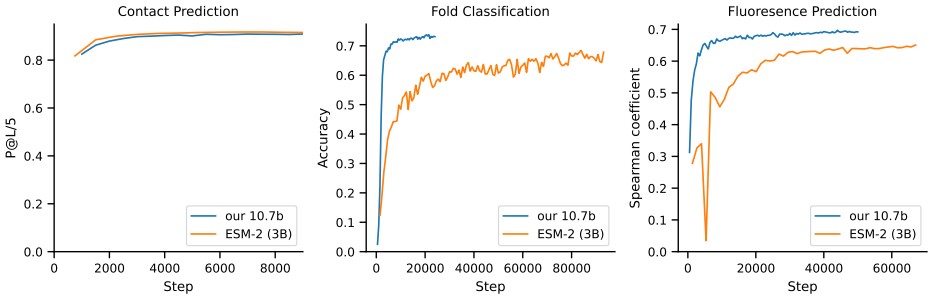

(a) **Learning Curve Convergence Speed.** LORA fine-tuning our 10.7B model and ESM-2 (3B) model on three downstream tasks.

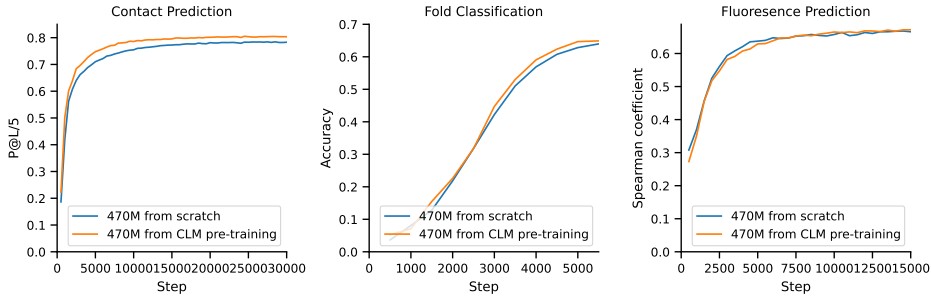

(b) **Learning Curve Convergence Rate Detection.**. LORA fine-tuning two 470M models on three downstream tasks. **transfer** means first pre-training 21B tokens on CLM then fine-tuning on MLM with 85B tokens, **from scratch** means training on 106B tokens from scratch.

# H    Mixed Objectives Training

We also employed an untied model to simultaneously optimize two objectives:

$$L_{CLM} = \text{CE}(V\sigma(W_1(\text{ encoder}(x))), y_{\text{next}}),$$

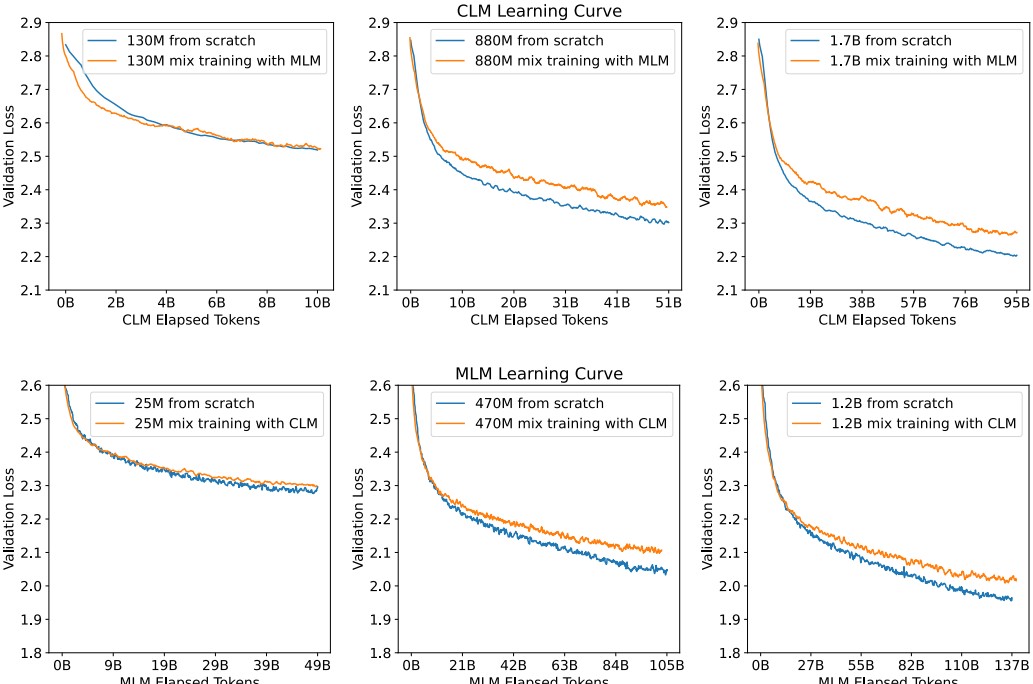

Figure A13: **Mixed objective validation loss**. Comparative validation loss curves for models trained from scratch versus mixed training approaches. Each panel corresponds to different model sizes, as indicated by the parameters. For each model, two training strategies were compared over an identical number of elapsed tokens: training from scratch (blue) and mixed training with the other objective (orange). Across all model sizes, training from scratch consistently achieves lower validation loss compared to mixed training, suggesting that mixed training may not be as effective as dedicated training for each individual objective.

and

$$L_{MLM} = \text{CE}(V\sigma(W_2(\text{encoder}(x))), y_{\text{mask}}),$$

where $V$ represents the protein vocabulary embedding, and $W_1$ and $W_2$ are the parameters corresponding to the CLM and MLM tasks, respectively. CE is the cross-entropy operator. The $\sigma$ is the Tanh activation function.

We compared CLM and MLM under our scaling law of optimal model and data size distributions. One approach involved training from scratch, while the other used mixed training. In the mixed training approach, the actual number of training tokens was higher due to the additional FLOPs consumed by another optimally trained objective, in other words. In other words, mixed training consumes the FLOPs of two optimal allocations; we only extracted the loss curve of one target for comparison. We extracted the loss curve of just one target for comparison with the from-scratch training. Our findings indicate that mixed training of the two targets can lead to detrimental interference, an effect not observable in smaller models, as depicted in Figure A13. As the model size increases to a hundred million or billion parameters, the differences become more pronounced. The possible reason for this situation is that mixed training has reduced the batch size for one of the objectives, making optimization difficult. We did not further investigate the impact of increasing the batch size and only observed based on the training tokens. However, we cannot rule out the possibility that they are mutually detrimental. Therefore, if both objectives are to be optimized concurrently, a sequential training strategy should be employed: first optimizing CLM, followed by MLM training. We consider that CLM is more challenging to predict than MLM, which may allow the model to capture more complex and implicit sequential features initially, thereby enhancing its ability to understand and predict masked words in subsequent MLM training.

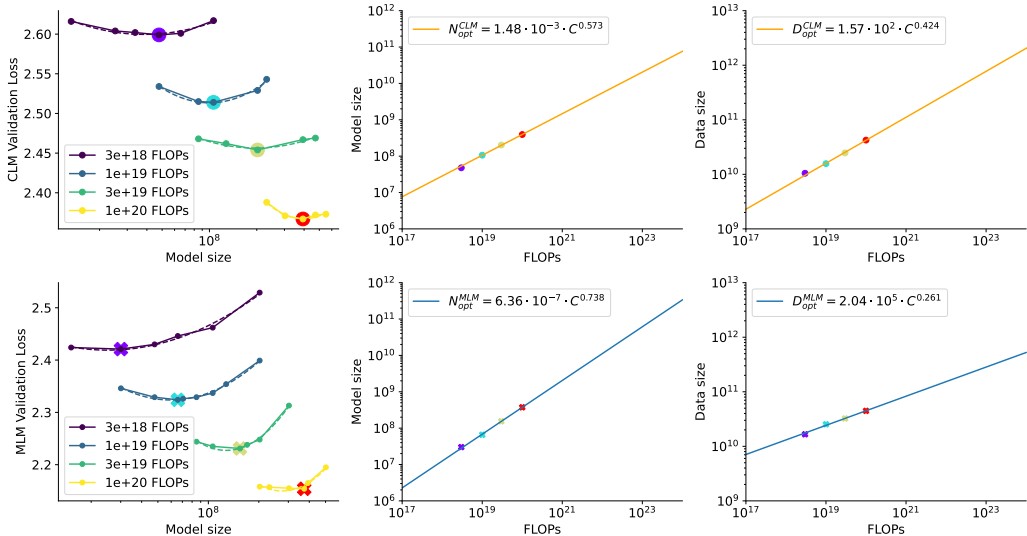

Figure A14: **Scaling laws of MoE.** The scaling behaviors of sparse parameter counts (8 experts) in MoE models, highlighting IsoFLOPs curves for different model sizes and FLOPs configurations. Each graph represents the relationship between model size, FLOPs, and validation loss for both CLM and MLM using MoE configurations. The power-law fits indicate optimal model size and data requirements for efficient scaling, showing that MoE models closely align with dense models in terms of scaling efficiency, with power-law coefficients for MoE-CLM and MoE-MLM approximating those of their dense counterparts. This suggests that MoE models can achieve similar scaling behaviors with potentially lower computational costs.

# I MoE Scaling

We find that the scaling behaviors of sparse parameter counts in Mixture of Expert (MoE) models are remarkably similar to those of dense model sizes, potentially allowing for a reduced compute budget for modeling scaling behaviors due to less activated parameters per token.

In our experiments, we evaluate MoE models ranging from 10M to 500M sparse parameter counts, using a model size of 17 with eight experts, following the settings outlined in Mixtral of experts [40], including its load-balancing scheme. The figure below shows different IsoFLOPs curves. Notably, the FLOPs here are calculated based on sparse parameters rather than actually activated ones. We use the method described in the main text to select optimal loss points and fit these around the sample points, enabling us to project the optimal model size and number of tokens for larger models (center and right). We observe that the power-law coefficients for CLM and MLM are similar to those of dense models, with MoE CLM vs. Dense CLM at approximately 0.57 vs. 0.58, and MoE MLM vs. Dense MLM at 0.74 vs. 0.77.

Our study strictly focuses on models with eight experts, which may not be entirely rigorous. Clark et al. [17] proposed a unified scaling law defining effective training parameters for MoE, aiming to harmonize the scaling laws for Dense and MoE models. Investigation of biological data will be considered as future work.

# J Combined Power-law

We applied the fitting function proposed by Chinchilla [36], detailed in Equation 8, to model the effects of various factors on model performance. It can provide a loss prediction where neither the parameters or model size are not optimal allocation. This loss function simultaneously depends on parameters $N$ and $D$:

$$L(N, D) = \frac{A}{N^\alpha} + \frac{B}{D^\beta} + E \tag{8}$$

where $E$ denotes the irreducible loss. Parameters $A$, $B$, $\alpha$, and $\beta$ are learned through the fitting process. As $N \to \infty$ or $D \to \infty$, the function degenerates to a form similar to Equation 2, which indicates that it models the scenarios under perfect conditions of other variables.

Given that most of our training tokens are used for less than or equal to one epoch, and that the model size is prone to underfitting at fixed FLOPs, the asymptotic behaviors $L(N)$ at $D \to \infty$ and $L(D)$ at $N \to \infty$ are enough for determining the parameters in $L(N, D)$.

To enrich data points, we randomly added several FLOP counts into 25% of the model size and trained these models for 0.25, 0.5, 0.75, and 1 epoch. And we adopt the Huber loss to fit these coefficients:

$$\min_{a,b,e,\alpha,\beta} \sum_{i} \text{Huber}_\delta \left( \text{LSE} \left( a - \alpha \log N_i, b - \beta \log D_i, e - \log L_i \right) \right), \tag{9}$$

where LSE represents the log-sum-exp operator, and $\delta = 10^{-3}$. The terms $N_i$, $D_i$, and $L_i$ denote the model size, dataset size, and loss of the $i$-th run, respectively. We fitted the MLM validation loss from 110 samples and the CLM validation loss from 149 samples using grid search with $\alpha \in \{0, 0.5, \ldots, 2\}$, $\beta \in \{0, 0.5, \ldots, 2\}$, $e \in \{-1, -0.5, \ldots, 1\}$, $a \in \{0, 5, \ldots, 25\}$, and $b \in \{0, 5, \ldots, 25\}$. The final initialized parameters of CLM and MLM both are $[e, a, b, \alpha, \beta] = [1, 5, 10, 0.5, 0.5]$. We set the maximum number of iterations to 1000, and the two objectives were essentially achieved after 360 iterations. The exponential powers of learned $a$ and $b$ yielded the coefficients $A$, $B$, which were reported in Table A9.

Table A9: Coefficient of Equation 8

| Objective | $A$ | $B$ | $\alpha$ | $\beta$ |
|---|---|---|---|---|
| CLM | 143.9 | 22036.5 | 0.367 | 0.496 |
| MLM | 3.365 | 7.569 | 0.042 | 0.099 |

Substituting all learned coefficients into the following Equation from the original Chinchilla paper:

$$N_{\text{opt}}(C) = G \left( \frac{C}{6} \right)^a, \quad D_{\text{opt}}(C) = G^{-1} \left( \frac{C}{6} \right)^b$$

$$where \quad G = \left( \frac{\alpha A}{\beta B} \right)^{\frac{1}{\alpha+\beta}}, \quad a = \frac{\beta}{\alpha + \beta}, \quad b = \frac{\alpha}{\alpha + \beta}. \tag{10}$$

The results closely approximate the trends given in Equations 1 and 2, confirming our overall findings.

## K  IsoLoss

In addition to using the seven different FLOPs counts reported in the main text to determine the optimal model sizes and fit our scaling law, we also incorporated additional model points into our analysis. We trained using the final loss points of all the CLM and MLM that are run. Figure A15 depicts the contour of the fitted function $L$ and the efficient frontier as a red dashed line, presented in log-log space. The frontier interval of Figure 2 is computed from this observation. From this approach, it revealed the scaling exponents for model size to be 0.77 in MLM and 0.57 in CLM, very similar to the IsoFLOPs profiling method in Section 3.1.

## L  Training Procedure

We conducted all experiments using Ampere A100 GPUs (80G) equipped with NVLink, utilizing the GLM framework [88, 26] developed based on DeepSpeed and Megatron. We have used a total of around 1 million GPU hours. Our approach predominantly utilized data parallelism, avoiding model parallelism and pipeline parallelism to simplify deployment. Modifications were made to the standard Transformer architecture [82], adopting a DeepNorm [85] strategy and layer normalization [5]. The activation function was set to GLU [71], RoPE [73] was used to encode position, similar to the settings found in the Transformer++ architecture [79]. We further adopt FlashAttention [18] to

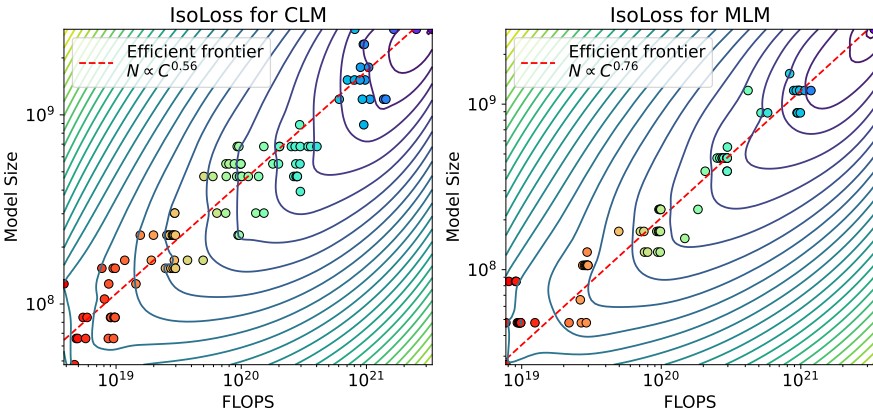

Figure A15: **Parametric fit for CLM and MLM**. Unlike the IsoFLOPs method used in the main text to select the optimal model size, these plots use all available data points to fit the models. The left panel shows the contour of the function $L$ and the efficient frontier (indicated by the red dashed line) for the CLM, and the right panel for the MLM. The rainbow dots represent identical loss. The results closely align with using the IsoFLOPs profiling method.

accelerate our training process. The used max LR empirically found to range between $6 \times 10^{-4}$ and $1.2 \times 10^{-4}$ from small to large model size, was used along with a cosine decay strategy to reduce it to $0.1 \times$ max LR. Both CLM and MLM were trained under similar settings for model size, with a consistent LR and a minimum warm-up period of 2.5% steps, extending to at least 100K training steps. All sequences were set to a length of 1024, with sequences concatenated using an <EOS> delimiter. Based on findings related to loss magnitude and batch size [53]. The AdamW optimizer [50] was used with $\beta_1 = 0.9$, $\beta_2 = 0.95$, $\epsilon = 1 \times 10^{-8}$, and a weight decay of $0.01$. All experiments omitted the dropout (it reduced the capacity to hinder model scaling) and trained with bfloat16. Most pre-training experiments were confined to the $\leq 1$ epoch, with some models extending up to 30% beyond one epoch. For the transfer learning setting, we load the finished checkpoint of the pre-training model and disregard the pre-trained optimized state, and learn rest tokens with warmup 5% steps the max LR.

## M  Broader Impact

If the scaling law of the protein language model improves predictions or understanding of protein structure and function, it could potentially have positive impacts on scientific research in fields such as biology, medicine, and drug development. This may facilitate the development of new drugs, accelerate progress in disease diagnosis, or drive advancements in frontier research in the life sciences.

## N  Model Parameters

Table A10 details the sizes and configurations of all models utilized in this research, training only with data parallel expcept 10B with tensor parallel size 2:

Table A10: **All model hyperparameters.** Several of the models presented have been trained using various learning rate schedules and differing amounts of training tokens.

| params | d_model | ffw | kv_size | head_num | layers |
|---|---|---|---|---|---|
| 4M | 192 | 512 | 24 | 8 | 8 |
| 5M | 256 | 683 | 32 | 8 | 7 |
| 6M | 256 | 683 | 32 | 8 | 8 |
| 10M | 320 | 853 | 40 | 8 | 8 |
| 13M | 320 | 1280 | 40 | 8 | 8 |
| 19M | 448 | 1194 | 64 | 7 | 8 |
| 25M | 512 | 1365 | 64 | 8 | 8 |
| 34M | 512 | 2048 | 64 | 8 | 8 |
| 40M | 576 | 1536 | 64 | 8 | 10 |
| 47M | 576 | 1536 | 64 | 9 | 12 |
| 66M | 640 | 2560 | 64 | 10 | 10 |
| 77M | 480 | 1280 | 24 | 20 | 28 |
| 85M | 768 | 2048 | 64 | 12 | 12 |
| 106M | 768 | 2048 | 48 | 16 | 15 |
| 127M | 768 | 2048 | 48 | 16 | 18 |
| 154M | 896 | 2389 | 64 | 14 | 16 |
| 157M | 640 | 1707 | 32 | 20 | 32 |
| 170M | 768 | 2048 | 48 | 16 | 24 |
| 200M | 896 | 2389 | 64 | 14 | 21 |
| 230M | 896 | 2389 | 64 | 14 | 24 |
| 300M | 1024 | 2731 | 64 | 16 | 24 |
| 393M | 1280 | 3413 | 80 | 16 | 20 |
| 470M | 1280 | 3413 | 80 | 16 | 24 |
| 550M | 1280 | 3413 | 80 | 16 | 28 |
| 670M | 1536 | 4096 | 96 | 16 | 24 |
| 880M | 1792 | 4778 | 64 | 28 | 23 |
| 1.2B | 2048 | 5461 | 64 | 32 | 24 |
| 1.5B | 2304 | 6144 | 64 | 36 | 24 |
| 1.7B | 2304 | 6144 | 64 | 36 | 28 |
| 2.0B | 2560 | 6832 | 64 | 40 | 26 |
| 2.4B | 2560 | 6832 | 64 | 40 | 30 |
| 2.8B | 2560 | 6832 | 64 | 40 | 36 |
| 3.1B | 2688 | 7168 | 64 | 42 | 36 |
| 3.4B | 2816 | 15040 | 128 | 22 | 22 |
| 4.0B | 3072 | 8192 | 128 | 24 | 36 |
| 5.7B | 3328 | 8874 | 128 | 26 | 40 |
| 6.2B | 3584 | 9556 | 128 | 28 | 40 |
| 7.2B | 4096 | 10923 | 128 | 36 | 36 |
| 10.7B | 4352 | 11605 | 136 | 32 | 47 |

