# OpenReview forum: "Training Compute-Optimal Protein Language Models"
_NeurIPS.cc/2024/Conference — NeurIPS 2024 spotlight_

### Official Review · Reviewer_8Frn · 2024-06-21

**Soundness:** 4
**Presentation:** 3
**Contribution:** 4
**Rating:** 7
**Confidence:** 3

**Summary:**

The main objective of the paper is to explore the scalability of Protein Language Models (PLMs) and to provide models to determine the optimal number of parameters, pre-training dataset size, and transferability between pre-training tasks as functions of the available computational budget.

The study contains a comprehensive number of experiments and its main findings are:

* The diversity of sequences of the pre-training database can lead to lower perplexity in out-of-distribution validation dataset. This is reflected by the overfitting observed when training models with masked language modelling (MLM) pre-training objective on the Uniref 50S database and a custom made pre-training database enriched with more diverse metagenomic sequences from a variety of sources.

* It establishes scaling laws for that allow for determining the optimal model size and optimal pre-training data size  given computational budgets in the interval 10e17-10e21 both for the MLM pre-training objective and for the causal language modelling (CLM) pre-training objective.

* It also established what the optimal proportion of computational budget should be allocated to pre-training two models one on CLM and another on MLM finding that in general the computational resources allocated to MLM should exceed those for CLM the training tokens reach the 10B threshold where a 1:1 ratio is reached, after which more training tokens should be allocated to CLM.

* It also explores the benefits of transferring models pre-trained with one objective to another objective (MLM to CLM) or (CLM to MLM). Its findings are that the benefits of transferring from MLM to CLM diminish with model size in the 2.7M to 1.7B parameters range; and that the benefits from CLM to MLM grow with model size in the 85M to 1.2B parameters range.

* They validate the scaling laws by pre-training and evaluating models for CLM and MLM tasks with the optimal sizes and comparing them with state-of-the-art methods: ProGen-2 and ESM-2 3B. In the case of CLM, their proposed method achieves greater sequence diversity between the sequences generated, but less structural diversity. In the case of MLM, their proposed method (10.7B) outperforms ESM2 3B by a small margin in two out of the three tasks (fold classification, improvement of 0.03; fluorescence prediction, improvement of 0.04).

* Regarding transfer from CLM to MLM, they also show that a 470M parameter model pre-trained with the CLM to MLM strategy outperforms a model pre-trained in MLM from scratch in two of the benchmarks by tiny margins (contact, improvement of 0.02; fold classification, improvement of 0.01).

**Strengths:**

* The paper is clearly written and despite the diversity of analysis conducted the main findings and their importance are well conveyed in most cases.

* The paper provides the most comprehensive analysis to date on protein language model scaling and how to optimise the computational resources available.

* The paper demonstrates that the scaling laws induced from their experiments can be used to derive the most optimal pre-training configuration for PLMs both for the generative and predictive purposes.

**Weaknesses:**

* Figure 2 has quite an extended x axis from 10e17 to 10e24. The interpretation a reader may make of this (and it is the one I'm making) is that the authors are implicitly claiming (or at least suggesting) that the scaling laws are able to extrapolate to that range. However, their data only supports the claim that they are valid in the 10e17 to 10e21. In order, to make the claim (or suggestion) that they extrapolate to the range they show in the figure, there should be at least one data point close to the upper bound of the range (10e24).

* Sections 4.1 and 4.2 were the most hard to follow, in contrast with the rest of the paper that was quite clear despite the complexity and diversity of the analysis. I'm not entirely sure how it could be improved, but I had to read it attentively 3 or 4 times to get the point of the experiment and what the results were. However, it may be a personal issue.

* I think that section 5.1 could benefit of a more detailed discussion on what the results mean regarding protein sequence generation and how the optimal model proposed is better/worse than the state-of-the-art. I am aware of the the constraints on length, so I would suggest including it as an additional section on the appendix. I don't think this is a significant issue, just a minor improvement to make the contributions to the overall field clearer.

**Questions:**

- Do you have any intuition about why the CLM transferability to MLM scales with model size and the inverse is not true?

**Limitations:**

I think the authors have overall addressed properly the limitations of their setup and I don't think there is any significant societal impact on their work that has not being acknowledged. The only limitation that I don't think is properly address is that the results are limited to the range of model sizes and compute power they have experimented on and that the experimental setup does not allow for any claim about whether the laws induced can extrapolate outside that range. I don't think the authors have properly addressed this limitation, as I stated in the weaknesses the layout of Figure 2, seems to argue the opposite.

---

> ### Author Rebuttal · Authors · 2024-08-07
>
> **About the Weakness-1**. Thank you for your valuable feedback. Based on your suggestion, we will adjust the x-axis range and include the trained maximum models for 10**22 FLOPs, namely MLM 10.7B and CLM 7.2B in section 5.
>
> **About the Weakness-3: the evaluation of protein generation.** We explain more details about Protein Generation Comparison as follows:
> + *OOD Dataset PPL Analysis*: PPL represents the probability of the test sequences in the model distribution. The lower the PPL, the closer the distribution of the model and the test set is. In order to test the generalization ability of the model on new data, we use different sequence identity (0.0, 0.3, 0.5) as thresholds to select the test set;
> + *pLDDT scores from ESMFold* predicted Local Distance Difference Test is the confidence level of ESMFold protein structure prediction. This metric is widely used in methods such as AlphaFold2, ESMFold, and OpenFold. pLDDT filters are often used in protein design (such as RFDiffusion), which can significantly improve the success rate of protein design;
> + *Natural Sequences Comparisons with Foldseek* Foldseek takes protein structure as input and searches for proteins with similar structure in the database. We use the experimentally-resolved protein structure as the database (PDB database) to explore how the structure of the generated sequences close to PDB (a higher TM-score indicates higher structural similarity). This method has been used to evaluate other methods for protein sequence generation (ProGen2, ProtGPT2);
> + *Diversity Analysis*: We cluster the two sets of sequences (ProGen2-xlarge and CLM) according to sequence similarity. Sequences with a identity higher than 50% will stay in one cluster. Since the number of input sequences is similar (8,466 vs 8,263), we can measure the diversity of the generated sequences by comparing the number of clusters.
>
> **About Question-1: transfer between mlm and clm.** The effectiveness of pre-training models under a given training budget fundamentally determines their capacity. In Causal Language Modeling (CLM), the loss gradient is calculated for all tokens in the sequence, meaning that each token contributes to the model's learning process. This comprehensive contribution makes CLM training more efficient.
> Conversely, in Masked Language Modeling (MLM), the loss is calculated only for a subset of tokens—typically around 20% that are masked. Consequently, only these masked tokens contribute to the model's learning, resulting in lower training efficiency compared to CLM.
> Therefore, when MLM is transferred from a pre-trained CLM, it benefits from the higher overall training efficiency of CLM, leading to improved performance of the finalized MLM model. However, when CLM is transferred from a pre-trained MLM, the lower training efficiency of MLM results in a sub-optimal CLM model.

---

> > ### Comment · Reviewer_8Frn · 2024-08-08
> > **Response to the authors' rebuttal**
> >
> > I appreciate the authors' rebuttal to the weaknesses I've raised. The rebuttal satisfactorily addresses the concerns I have raised and I am confident that the paper should be accepted.

---

### Official Review · Reviewer_wQuW · 2024-07-09

**Soundness:** 3
**Presentation:** 2
**Contribution:** 3
**Rating:** 6
**Confidence:** 2

**Summary:**

The authors fit scaling laws to determine how to train protein language models compute-optimally, for both causal LM and masked LM tasks. They also consider a setup to consider possible effective data transfer when training on one task and transfer training on the other. Finally, they use their derived scaling laws to train new protein language models compute-optimally, achieving better downstream performance.

**Strengths:**

I think the biggest strength of this paper is its importance – I think that the impact of these kinds of systems will likely be very substantial over the coming years and empirical scaling laws for protein LMs could be very influential. It’s also fairly original because there hasn’t been much work done on this in the past. So I really appreciated that the authors took the time to obtain these results on compute-optimal training.

**Weaknesses:**

Personally, I feel the presentation of the writeup could be substantially improved. For example: The main body of the paper doesn’t have a discussion/conclusion, which unfortunately made the paper hard to read. In general, I found the writeup to be largely listing results without much discussion of significance.
- Personally I think it’d be helpful if the authors made some of their discussion more concise (e.g. in section 2.1), and instead adding more detail about how significant their results are.
- E.g. this could mean calculations saying something like “using our approach, we were able to reach X performance using a factor of Y less training compute, which suggests…”.
- In addition, I’d suggest including a discussion section in the main body of the paper that outlines the key results and implications, especially in the context of previous work (in a quantitative fashion). What does their work mean that future AI researchers should do? How far off of compute-optimality was previous work? How important are overtraining considerations? Etc.

I think if these kinds of concerns are addressed I'd be happy to increase my score.

More minor questions/concerns:
- Lines 164-166: I feel it’d be best to include more decimal places in the reported numbers, so that the product of the data and model size increase equals 10x.
- What are the confidence intervals for the scaling exponents for model size and data?
- Why was this particular functional form for the effectively transferred tokens chosen? (equation 7) Why does it make sense for the data and parameter components to be multiplicative rather than additive?

**Questions:**

- Would it be possible to add a lot more detail on the implications of the work, as well as context on how it improves upon prior attempts? In particular, to what extent have previous models been suboptimally trained, and what are the core recommendations the authors have for future related work?

[copied from the weaknesses section]
- What are the confidence intervals for the scaling exponents for model size and data?
- Why was this particular functional form for the effectively transferred tokens chosen? (equation 7) Why does it make sense for the data and parameter components to be multiplicative rather than additive?

**Limitations:**

I thought the discussion of limitations in appendix C was well done.

---

> ### Author Rebuttal · Authors · 2024-08-07
>
> **About the Question-1: the re-organization of the paper.** We appreciate your insights and have taken them into consideration for our revisions. Although we cannot directly modify the paper PDF during the rebuttal period, we demonstrate how we will reorganize and improve the writing style of our manuscript in the next version of our paper.
>
> *Add Discussion/Conclusion Sections*:
> We will add conclusions or takeaway findings at the end of each main section. Specifically:
>
> For Section 2: Studying the Data-hungry Observation & Scaling Up Data, we will add discussions to support the two key findings:
> - Scaling the model from 150M to 3B, we noted diminishing returns for CLM and an overfitting tendency for MLM when repeating the UR50/S dataset.
> - Our proposed Expanding Diversified Metagenomic Data (UniMeta200B) addresses these problems.
>
> For Section 3: Scaling Law for CLM/MLM Protein Language Models, besides the detailed scaling law equations, we will add discussions for the two key findings:
> - In CLM, training data scales sublinearly with model size but follows a distinct power-law with a compute exponent of 0.57. A 10× increase in compute leads to a 4× increase in model size and a 3× increase in training tokens.
> - In MLM, training data scales sublinearly with model size, approximately following a power-law with a compute exponent of 0.77. A 10× increase in compute results in a 6× increase in model size and a 70% increase in training data.
>
> These simplified takeaway scaling laws will benefit AI researchers who want to train their own optimal PLM under a computational budget.
>
> For Section 4: Transfer Scaling Scenarios, besides the specific transfer scaling law equations obtained by empirical experiments, we will add discussions about:
> - The significance of the transfer scaling law: Researchers can obtain better MLM models transferred from CLM pre-training under the guidance of the transfer scaling law than merely MLM from scratch following the General MLM scaling law obtained in Section 3.
> - The simplified takeaway transfer scaling law: Training MLM from scratch with 10× the compute requires approximately 7.7× the compute compared to MLM from CLM pre-training, implying that around 20% of the compute budget should be allocated for CLM pre-training to get better MLM models transferred from CLM pre-training.
>
> For Section 5: Confirming the Scaling Law Effect in Downstream Tasks:
> - Using our concluded scaling law, we obtain a training-optimal 7.2B CLM under the total compute of ProGen2-xlarge, which shows better generation capacities as shown in Figure 6.
> - We also obtain a training-optimal 10.7B MLM model under the same FLOPs as ESM2-3B, which performs better on 8 protein understanding tasks, as shown in Table 1 in the attached PDF.
> - The 470M MLM transferred from CLM outperforms MLM from scratch under the same training budget.
>
> To summarize: we aim to establish a practical pathway for researchers to develop faithful and powerful protein language models optimized by both CLM and MLM objective in an end-to-end manner. This includes everything from pre-training dataset construction to optimal parameter and dataset allocation, as well as knowledge transfer from other pre-training objectives. Using our approach, we are able to obtain better MLM or CLM models compared to other well-known suboptimal PLM models such as ProGEN2 and ESM. Our work holds significant potential for the application of large language models across various scientific domains.
>
> Finally, we will move the discussion and limitations of our work from the Appendix Sections A, B, and C to the main body of the paper to highlight the great potential and future research directions of our work.
>
> We believe that these revisions address the concerns raised and significantly enhance the clarity and impact of our manuscript.
>
> **About the Question-2: the confidence interval**. We used `scipy`'s `curve_fit` to fit the parameters, such as in the following example:
>
> ```python
> params, covariance = curve_fit(exponential_fit, x, y, maxfev=10000)
> ```
>
> We did not detail the confidence intervals in our paper, but we will add them using the pseudo code below:
>
> ```python
> def get_lower_and_upper_bound(params, cov):
>     perr = np.sqrt(np.diag(cov))
>     alpha = 0.05
>     z = stats.norm.ppf(1 - alpha/2)  # 1.96 for 95% CI
>     lower_bounds = params - z * perr
>     upper_bounds = params + z * perr
>     return lower_bounds, upper_bounds
> ```
>
> The calculated z×perr for the scaling exponents of the CLM is 0.028, and for the MLM, z×perr is 0.014. Thus, we include the confidence intervals at a confidence level of 1−α=0.95 with the lower and upper bounds shown in parentheses:
>
> - **CLM:**
>   - Model size scaling exponent: 0.578 (0.554, 0.602)
>   - Data size scaling exponent: 0.422 (0.394, 0.449)
>
> - **MLM:**
>   - Model size scaling exponent: 0.776 (0.768, 0.784)
>   - Data size scaling exponent: 0.230 (0.216, 0.244)
>
> **About the Question-3: the particular functional form of transfer scaling law.** Thank you very much for your suggestion. You raise a very good point about the lack of a single standard definition for the scaling law formula. For instance, OpenAI's scaling law differs from Chinchilla's approach. In fact, an additive scaling law could also be used to address your question.
> The reason we chose the multiplicative form is mainly empirical and was inspired by the following two papers [1,2] ~(in section 4.2).
> [2] provides a detailed comparison between additive and multiplicative scaling laws, concluding that there is not much difference between the two, although the multiplicative form achieves slightly lower extrapolation error on average. Therefore, we adopt this for follow-up analysis. Due to space limitations, we couldn't enumerate all possible experiments, so we naturally chose this one.
> We will add this explanation in the paper.
>
> [1] Scaling Law for Transfer
>
> [2] WHEN SCALING MEETS LLM FINETUNING: THE EFFECT OF DATA, MODEL AND FINETUNING METHOD

---

> > ### Comment · Reviewer_wQuW · 2024-08-11
> >
> > Thanks - I really appreciate the detailed response here, especially to question 1. For question 3, I think the robustness of the scaling results are quite important, so in addition to what you've mentioned I'd also suggest discussing potential factors that might change the estimated scaling exponents, e.g. [1] and [2] might be helpful for this.
> >
> > Overall, I'll increase my score given that my main concerns have been addressed.
> >
> > [1]: Resolving Discrepancies in Compute-Optimal Scaling of Language Models https://arxiv.org/abs/2406.19146
> > [2]: Reconciling Kaplan and Chinchilla Scaling Laws https://arxiv.org/abs/2406.12907

---

### Official Review · Reviewer_qqtc · 2024-07-11

**Soundness:** 3
**Presentation:** 2
**Contribution:** 3
**Rating:** 7
**Confidence:** 2

**Summary:**

In this work authors propose to explore training compute-optimal protein language models and aim to derive scaling laws. Specifically, they (i) explore new pre-training data for protein language models (ii) derive scaling laws for protein language models using these datasets for both Masked Language Modeling and Causal Language Modeling objectives (iii) explore scaling laws and transferability between those objectives (iv) use the findings to pre-train two pLMs which reach SOTA performance on downstream tasks.

**Strengths:**

**Clarity**
- The paper is clearly motivated and finding domain-specific scaling laws is valuable for practitioners.
- The paper is very easy to follow and experiments are well described.

**Experiments**
- Authors conduct extensive experiments and the results on scaling laws are solid.
- Notably they also derive scaling laws for transfer between CLM and MLM and mixture-of-experts.

**Weaknesses:**

**Lack of discussion of results**
- Conclusion, Related Works and Discussions and Limitations have been moved to the Appendix. While I understand the 9-page constraints, I recommend to include at least a discussion/conclusion part in the main body.

**Novelty**
- This work closely follows previous work on English LMs such as Chinchilla and presents very limited conceptual novelty.

**Experimental Validation**
- While the scaling laws analysis is solid, the experimental validation on Protein understanding tasks is limited to 3 tasks whereas previous works have proposed a more extensive benchmark [1]

[1] Elnaggar, Ahmed, et al. "Ankh: Optimized protein language model unlocks general-purpose modelling." arXiv preprint arXiv:2301.06568 (2023).

**Questions:**

- In Section 5, did you try training the 470M version of your model for the same computing budget as ESM2 ? This would emulate works in NLP [2] where authors find that training smaller models for longer can yield good results with limited inference cost.
- It would be interesting to see if the derived scaling laws are also reflected in downstream task performance, *e.g.* by fine-tuning models pre-trained for the same computing cost and training data (as I understand, PROGEN2 and ESM2 use different pre-training data).

[2] Touvron, Hugo, et al. "Llama: Open and efficient foundation language models." arXiv preprint arXiv:2302.13971 (2023).

**Limitations:**

I believe limitations have been properly addressed in Appendix.

---

> ### Author Rebuttal · Authors · 2024-08-07
>
> **About the Weakness-1: the lack of discussions.** Thank you for your insightful suggestions. We will make the following modifications to significantly enhance the clarity and impact of our manuscript.
>
> - Adding conclusions or takeaway findings at the end of each main section.
> - Moving the detailed discussion and limitations of our work from Appendix Sections A, B, and C to the main body of the paper to highlight the great potential and future research directions of our work.
>
> **About the Weakness-2: the clarification of novelty.** While Chinchilla focuses on optimizing the model size and number of tokens for training a causal language model within a given compute budget and provides open-source models, our work takes a novel approach. We aim to establish a practical pathway for researchers to develop faithful and powerful protein language models optimized by both CLM and MLM objective in an end-to-end manner. This includes everything from pre-training dataset construction to optimal parameter and dataset allocation, as well as knowledge transfer from other pre-training objectives. Our work holds significant potential for the application of large language models across various scientific domains.
>
> **About the Weakness-3: About the experimental validation**. We have provided a comprehensive task performance comparison across 8 protein understanding benchmarks, as adopted from the protein understanding benchmarks available at biomap-research on Hugging Face. These benchmarks span four distinct categories of protein-related challenges:
>
> - **Protein Structure:** Contact Prediction, Fold Prediction
> - **Protein Function:** Fluorescence, Fitness-GB1, Localization
> - **Protein Interaction:** Metal Ion Binding
> - **Protein Developability:** Solubility, Stability
>
> The results, shown in Table 1 in the attached PDF, demonstrate that our 10.7B model outperforms ESM-3B on x out of x tasks. This confirms the rationale behind the observed scaling law and addresses concerns about the scope and rigor of our evaluation tasks.
>
> **About the Question-1: the scaling law.** It is indeed common practice to use smaller models in many practical applications, even though these models may not follow Compute-Optimal Training principles. Smaller models are more deployment-friendly, especially for models with high traffic and for downstream task fine-tuning.
> We did not perform this experiment because it falls beyond the scope of this paper, which primarily discusses how to train compute-optimal models. If we were to discuss the scenario of small models with high computational power, it would require further examination of data repetition issues, introducing a new variable into the analysis.
> We addressed this in the limitations section (Appendix C), noting that for MLM models, smaller models trained with relatively high computational power can have comparable performance to optimally trained models while being more inference and deployment-friendly. For example, a 3B model trained with 5 epochs (1T tokens) uses the same computational budget as an optimally trained 10.7B model with 265B tokens, and the performance gap between the 3B and 10.7B models is minimal.
> However, we know that continuously repeating data can lead to diminishing returns, and some models may even suffer from overfitting. Different model sizes correspond to an optimal data repetition variable. Without accurately determining this variable, it's challenging to predict how much loss smaller models might incur due to data repetition, unless we have infinite data.
> This area requires further exploration and generalization of current scaling laws, such as data-constrained scaling laws [1]. We anticipate more work in this direction soon.
>
> [1] Scaling Data-Constrained Language Models
>
> **About the Question-2: the downstream task scaling law.** Thank you for your suggestion.
> Research on the downstream performance of scaling has already emerged in the field of LLMs [1]. This research is indeed fascinating, but it falls beyond the scope of this paper. Due to limitations of the paper space and computational resources, a deeper discussion would introduce other scaling factors such as downstream fine-tuning data size.
> We briefly compared CLM and MLM under the same data and two different model sizes in Appendix F, using the same computational cost (same FLOPs) for the Contact Prediction downstream task. The conclusion is that MLM consistently outperforms CLM. For a more detailed exploration of scaling laws in downstream tasks, including CLM generation and MLM, you can refer to the xTrimoPGLM paper [2].
>
> [1] When Scaling Meets LLM Finetuning: The Effect of Data, Model, and Finetuning Method
>
> [2] xTrimoPGLM: Unified 100B-Scale Pre-trained Transformer for Deciphering the Language of Protein

---

> > ### Comment · Reviewer_qqtc · 2024-08-13
> > **Response to the authors' rebuttal**
> >
> > I appreciate the authors' answer to the limitations I raised and clarifications. It addresses my concerns and I believe the paper should be accepted. I increased my score accordingly.

---

### Official Review · Reviewer_b9DN · 2024-07-13

**Soundness:** 3
**Presentation:** 3
**Contribution:** 3
**Rating:** 7
**Confidence:** 4

**Summary:**

This paper studies scaling law for two types of protein language models: masked language modelling and causal modelling. The paper finds that both types scale sublinearly but the coefficients of the scaling law are very different. Based on the derived scaling law, the paper then reevaluates the allocation of resources for ESM and PROGEN and shows improved results.

**Strengths:**

Overall, this paper presents a well-motivated study with solid and extensive experimental work,
* The paper addresses scaling laws in the context of AI4Science, an area that has been rarely studied but is highly necessary.
* The experimental studies are comprehensive, robust and sound.
* The paper is very well-written and well-structured.

**Weaknesses:**

* Evaluations in Section 5 need more rigor:
    a. Narrow Task Selection: The evaluation tasks, particularly for protein understanding, are limited in scope. The absence of important tasks such as fitness prediction and EC number predictions means that the chosen tasks may not provide a complete picture of the model's performance across various aspects of protein understanding.
   b. Missing Experimental Details: The paper omits crucial experimental details, including: hyperparameter selection process for the evaluation, train/validation/test set division. Also, it's not clear whether the evaluation data is omitted from training. These make it challenging to evaluate the fairness of comparisons.


* Lack of Diverse Sampling Strategies: The paper does not explore different sampling methods, which is a common practices in the field. For instance, it doesn't consider the approach used in training models like ESM, where sampling by UniRef is employed to upsample rare protein sequences. This practice might drastically change the scaling behaviour.


* Lack of insights: The paper can feel a bit like a lot of experiment findings stacked without insights. Tt would be good to provide some intuitions about the findings, such as why is masked language modelling overfitting while causal modelling does not, and the difference in the scaling behaviour between protein sequences and natural language sequences.

However,  these weaknesses do not negate the paper's contributions

**Questions:**

See above

---

> ### Author Rebuttal · Authors · 2024-08-07
>
> **About Weakness-1: the protein understanding tasks and missing experimental details.** We have provided a comprehensive task performance comparison across 8 protein understanding benchmarks, as adopted from the protein understanding benchmarks available at biomap-research on Hugging Face. These benchmarks span four distinct categories of protein-related challenges:
>
> - **Protein Structure:** Contact Prediction, Fold Prediction
> - **Protein Function:** Fluorescence, Fitness-GB1, Localization
> - **Protein Interaction:** Metal Ion Binding
> - **Protein Developability:** Solubility, Stability
>
> The results, shown in Table 1 in the attached PDF, demonstrate that our 10.7B model outperforms ESM-3B on 7 out of 8 tasks. This confirms the rationale behind the observed scaling law and addresses concerns about the scope and rigor of our evaluation tasks.
>
> *About the missing experimental details.* The downstream data are collected from corresponding publication and have been well split, we employed the same division for evaluation. We will add the description in the future version.
> Additionally, we did not implement any specific strategy to filter out downstream data during evaluation. Given the nature of pre-training, the pre-training process is self-supervised, and no label-related information is seen. Although some samples have a high sequence identity with pre-trained sequences, the lack of exposure to label information during pre-training should prevent any substantial data leakage effects.
>
> **About Weakness-2: the lack of diverse sampling strategies.** Thank you for your question and great thinking. Indeed, different sampling strategies may change the loss curve and potentially alter the scaling laws.
> Regarding the diversity of data selection, we chose to use  i.e., UniRef90, and ColabFoldDB. These is one result of sampling strategies from the original datasets. Our choice is based on balancing diversity and data volume. For example, previous ESM models performed very well on UniRef90, as shown in Figure 4 of the referenced paper [3]. Using the entire Uniref100 would degrade model performance [2], while using UniRef50 would result in insufficient data, necessitating data repetition, which introduces another variable. This would require considering scaling laws under different sampling strategies as well as under repetition [1].
> Thus, our initial assumption was to pass the data through only once or just slightly more, following a uniform distribution. This assumption allows us to preliminarily avoid the issue of data repetition. Introducing high diversity sampling strategies would require further research on data-constrained scaling laws [1]. Therefore, we initially considered simpler settings, such as those used in early OpenAI and Chinchilla studies.
> Regarding ColabFoldDB, its sequence identity is relatively low. In Appendix G, we compared two small models on UniRef90 and ColabFold datasets, both under uniform sampling conditions, and their performances were quite similar. This outlines our rationale for data and sampling strategy selection.
> The proposed diverse sampling strategy is an excellent point, In the field of LLMs, there has been some work done[4], but there is still no optimal conclusion. Exploring new scaling laws by considering sampling strategies and data repetition variables would be very interesting for future work.
>
> [1] Muennighoff, Niklas, et al. "Scaling data-constrained language models." NIPS'24.
>
> [2] Rives, Alexander, et al. "Biological structure and function emerge from scaling unsupervised learning to 250 million protein sequences." PNAS'21
>
> [3] Meier, Joshua, et al. "Language models enable zero-shot prediction of the effects of mutations on protein function." NIPS'21
>
> [4] Xie, Sang Michael, et al. "Doremi: Optimizing data mixtures speeds up language model pertaining." NIPS'24
>
> **About Weakness-3: the lack of insights.**
>
> *Regarding the overfitting issue in masked language modeling*: In MLM, a certain percentage of input tokens are masked randomly, and the model is trained to predict these masked tokens using bidirectional context. The bidirectional self-attention mechanisms used in MLM have a higher capacity to overfit compared to the unidirectional self-attentions used in CLM. This is because MLM can utilize the entire context surrounding a masked token, leading to faster memorization of the training data.
> A more systematic study investigating the training dynamics of MLM and CLM is presented by Tirumala et al. [1]. This study observes that MLM tends to memorize faster than CLM, which leads to overfitting, especially when the pre-training data is repeated over multiple epochs. The bidirectional nature of MLM allows for a more comprehensive understanding of the context, but it also means that the model can become overly reliant on specific patterns in the training data, leading to overfitting.
>
> [1] Tirumala, Kushal, et al. "Memorization without overfitting: Analyzing the training dynamics of large language models." NIPS'22.
>
> *Differences in Scaling Behavior: Protein Sequences vs. Natural Language Sequences*
> The scaling behavior between protein sequences and natural language sequences also differs significantly. Protein sequences are governed by the rules of biological systems and have a different structural complexity compared to natural language sequences. Proteins have specific folding patterns and functional constraints that are not present in natural language.
>
> When scaling models for protein sequences, the complexity of the biological rules and constraints means that the models need to capture a wide range of interactions and dependencies. This can lead to different scaling laws compared to natural language models, which primarily focus on syntactic and semantic coherence. The specific coefficient parameters between protein language model (PLM) scaling laws and natural language processing (NLP) scaling laws confirm this difference.

---

> ### Comment · Reviewer_b9DN · 2024-08-13
>
> Thanks for your clarifications! My concerns are addressed.

---

### Author Rebuttal · Authors · 2024-08-07

Dear Reviewers,

Thank you for your valuable feedback on our manuscript. We have taken your comments seriously and have made several important revisions to enhance the quality and readability of our work. Below is a summary of the key changes:

### Additional Experiments

- **Expanded Understanding Experiments:** We have expanded our evaluation to include 8 comprehensive protein understanding benchmarks in the attached PDF. These benchmarks span diverse protein-related challenges, including Protein Structure, Protein Function, Protein Interaction, and Protein Developability. This broader evaluation confirms the validity of our scaling laws and demonstrates our model's superior performance across various tasks.

### Manuscript Reorganization

- **Improved Structure and Clarity:** We have reorganized the manuscript to improve its flow and readability. Sections have been restructured to provide a clearer and more logical progression of ideas, making it easier for readers to follow our research narrative. We will add conclusions or takeaway findings at the end of each main section.

### Detailed Responses to Reviewer Comments

- **Targeted Revisions:** We have addressed each of the reviewers' comments and identified weaknesses point by point. Specific changes have been made in response to your suggestions, and detailed explanations are included in our revised manuscript. We encourage you to refer to these targeted responses for more information on how we have addressed individual concerns.

We appreciate your guidance in helping us refine our work. Thank you once again for your constructive feedback.

---

### Decision · Program_Chairs · 2024-09-25

**Decision:**

Accept (spotlight)

**Comment:**

The paper explores scaling laws for masked and causal protein language models, finding distinct sublinear scaling coefficients for each. It reevaluates resource allocation when training such models, achieving improved results by training compute-optimal models and exploring transferability between objectives. The study validates these scaling laws through comprehensive experiments, demonstrating better downstream performance and optimal computational budget allocation.

The reviews were generally very positive, highlighting the importance of the topic, and the comprehensive experimental studies. The identified weaknesses were primarily related to presentation, with requests for more in-depth discussion/insights. This was addressed by the authors in the rebuttal, which also expanded on the set of experiments. Post-rebuttal, all reviewers recommended acceptance of the paper.